# AN EXPLORATION WITH ENTROPY CONSTRAINED 3D GAUSSIANS FOR 2D VIDEO COMPRESSION

**Xiang Liu**[1,3]   **Bin Chen**[2,3*]   **Zimo Liu**[3]   **Yaowei Wang**[2,3]   **Shu-Tao Xia**[1,3]
[1]Tsinghua Shenzhen International Graduate School, Tsinghua University
[2]Harbin Institute of Technology, Shenzhen   [3]Peng Cheng Laboratory
`liuxiang22@mails.tsinghua.edu.cn, chenbin2021@hit.edu.cn`
`liuzm@pcl.ac.cn, wangyw@pcl.ac.cn, xiast@sz.tsinghua.edu.cn`

## ABSTRACT

3D Gaussian Splatting (3DGS) has witnessed its rapid development in novel view synthesis, which attains high quality reconstruction and real-time rendering. At the same time, there is still a gap before implicit neural representation (INR) can become a practical compressor due to the lack of stream decoding and real-time frame reconstruction on consumer-grade hardware. It remains a question whether the fast rendering and partial parameter decoding characteristics of 3DGS are applicable to video compression. To address these challenges, we propose a Toast-like Sliding Window (TSW) orthographic projection for converting any 3D Gaussian model into a video representation model. This method efficiently represents video by leveraging temporal redundancy through a sliding window approach. Additionally, the converted model is inherently stream-decodable and offers a higher rendering frame rate compared to INR methods. Building on TSW, we introduce an end-to-end trainable video compression method, GSVC, which employs deformable Gaussian representations and optical flow guidance to capture dynamic content in videos. Experimental results demonstrate that our method effectively transforms a 3D Gaussian model into a practical video compressor. GSVC further achieves better rate-distortion performance than NeRV on the UVG dataset, while achieving higher frame reconstruction speed (+30% fps) and stream decoding. Code is available at Github.

## 1 INTRODUCTION

Video compression is a classical issue in the domain of signal processing. With the growing amount of video data on the Internet, numerous video compression algorithms have emerged. Classic video compressors employ hand-crafted rules to reduce redundant information in the data (Le Gall, 1991; Wiegand et al., 2003; Sullivan et al., 2012). Learning-based approaches, on the other hand, learn the characteristics of video from a large quantity of video data through a data-driven manner and achieve highly efficient compression (Lu et al., 2019; Shi et al., 2022; Li et al., 2024). Besides, methods based on implicit neural representation (INR) have also started to be applied in video compression, which represent signals by directly mapping coordinates or timestamps to the corresponding RGB values, offering new insights for the field (Chen et al., 2021; Lee et al., 2023).

Similar to the success of deep learning in other fields (Huang et al., 2023; Qin et al., 2024; Xia et al., 2024; Gao et al., 2024), current learning-based neural video compression approaches have achieved considerable progress in rate-distortion (RD) performance. However, these methods do have certain issues. Firstly, many methods have a high decoding computational complexity, limiting their practicality in real-world scenarios. INR-based method have to some extent alleviated this problem (Chen et al., 2021), but for those with superior RD performance, their rendering frame rates cannot reach the speed for practical usage (Kwan et al., 2023). Secondly, fully decoding the representation network is necessary before rendering any frame for INR-based method, making it challenging to achieve stream decoding. Given that neural representation is more suitable for

---

*Corresponding author

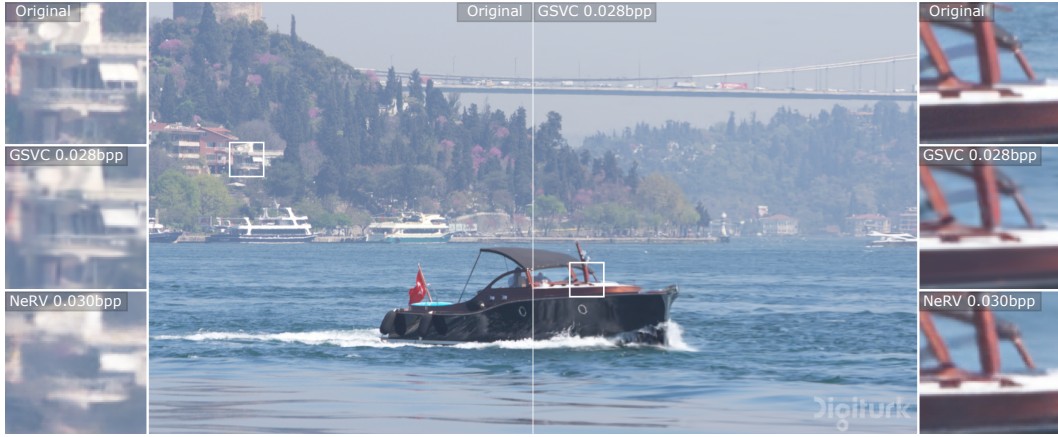

Figure 1: Comparison between proposed GSVC and NeRV (Chen et al., 2021). Our method achieves effective video compression while surpassing NeRV in some aspects such as background texture.

encoding longer videos (Chen et al., 2021), this characteristic significantly limits its applications, especially in popular streaming media.

Recently, 3D Gaussian Splatting (3DGS) (Kerbl et al., 2023) has gained increasing attention for its impressive capability of reconstructing 3D scene from multiple photos, which outperforms NeRF (Mildenhall et al., 2020) in training and rendering speed. Although 3D Gaussian representation is a method specifically designed for 3D scenes, it shares similarities with neural representation in that they are both *a new type of signal representation* and have the potential to be a general-purpose representation method applied to other domains (Dupont et al., 2021; 2022; Zhang et al., 2024). These fields will also benefit from the exclusive features of 3DGS, such as explicit representation, fast rendering, and partial parameter rendering.

In this paper, focusing on the weakness of INR video compressors and advantages of 3DGS mentioned before, we explore the feasibility of 3DGS in video compression. We propose a novel regional orthographic projection called Toast-like sliding window (TSW) orthographic projection. TSW orthographic projection utilizes the redundancy between adjacent frames without requiring any explicit or implicit 3D structure in the signal. Using the method, any model sharing same rendering process with 3DGS can be easily extended to a 2D video representation model. Based on TSW, we propose an end-to-end trainable Gaussian Splatting Video Compressor (GSVC) by extending the framework of HAC (Chen et al., 2024). Furthermore, we introduce time-aware Gaussians generation, which offers increased parameter efficiency. This design, cooperated with proposed optical flow (Horn & Schunck, 1981) guided deformable Guassians (Yang et al., 2024c), significantly improve the capability of GSVC to efficiently model dynamic objects in video. Fig. 1 demonstrates a qualitative results of GSVC.

The proposed TSW has been tested with vanilla 3DGS(Kerbl et al., 2023), Scaffold-GS (Lu et al., 2024) and HAC (Chen et al., 2024) to validate its universality, demonstrating that our method effectively transforms a 3D Gaussian model into a practical video compressor. Thorough experiments over UVG dataset (Mercat et al., 2020) were performed, in which proposed GSVC demonstrates better RD performance than NeRV (Chen et al., 2021) and superior rendering efficiency. Besides, our method achieves stream decoding by natural.

The primary contributions of this work are summarized below:

- We propose a novel TSW orthographic projection, which bridging the gap between 3D Gasussian Splatting and 2D video representation.

- We build an end-to-end trainable video codec GSVC based on TSW by extending an existing 3DGS compressor.

- We refine GSVC by time-aware Gaussian generation and optical flow guided deformable 3D Gaussian to improve parameter efficiency and capture dynamic contents in videos.

## 2 RELATED WORK

### 2.1 LEARNED VIDEO COMPRESSION

Traditional video codecs utilize a variety of hand-crafted rules to eliminate spatial and temporal redundancies such as different intra-prediction and inter-prediction patterns in order to achieve high-efficiency encoding (Le Gall, 1991; Wiegand et al., 2003; Sullivan et al., 2012). In video compression methods using deep learning, one approach is to improve existing traditional methods by replacing certain components, such as entropy coding (Song et al., 2017) and post-processing (Lu et al., 2018; Song et al., 2018) with neural network. Tian et al. (2024) propose a machine-friendly video compressor with traditional-neural mixed coding framework.

Another approach is to build an end-to-end deep learning video codec which jointly optimizes all the components in video compression pipeline (Lu et al., 2019; Hu et al., 2021). Such methods typically take advantage of representation ability of neural networks to achieve functions such as feature transformation and motion compensation (Lin et al., 2020). Liu et al. (2020) utilize the conditional entropy between frames. Li et al. (2021) propose using feature domain context as condition.

Besides typical end-to-end models, INR-based compression methods, which general require lower computational resources, provide a different view of compression task. These representation methods encode the signal itself implicitly in the weights of a neural network, The neural representation has attracted attention for its performance in synthesizing new views in 3D scenes (Mildenhall et al., 2020), and it has gradually been applied to the representation and compression of images and videos (Dupont et al., 2021; 2022; Ladune et al., 2023). Chen et al. (2021) first propose to use timestamps as the input of the representation network, outputting corresponding video frames. Combining with network pruning and quantization, this representation method can more effectively capture intra-frame and inter-frame relationships than NeRF (Mildenhall et al., 2020) in video compression. Subsequent work built on this foundation, with more detailed design of the network structure, achieving better rate-distortion performance (Chen et al., 2021; 2023; Lee et al., 2023; Kwan et al., 2023).

In comparison to end-to-end video compression models, INR-based methods can achieve lower computational overhead while not requiring pre-trained models to be deployed at the encoding and decoding ends. However, neural representation-based video compression methods still have difficulty meeting the real-time characteristics in rendering rate, especially for some models with better RD performance. In addition, these methods need to fully decompress the entire model for frame reconstruction, making it impractical to apply to scenarios where streaming transmission is required.

### 2.2 3D GAUSSIAN SPLATTING

3D Gaussian Splatting (3DGS) (Kerbl et al., 2023) is a method for 3D scene reconstruction that has emerged in recent years, which has attracted the attention of many researchers with its fast reconstruction speed and real-time rendering characteristics. Different from implicit representation of NeRF, 3DGS uses a series of Gaussian points in space to represent the 3D information of the scene. This representation method also provides a new perspective for many tasks. For example, 4D dynamic videos can be represented through moving and deforming Gaussian points in a deformation field (Lin et al., 2024; Yang et al., 2024c). Shi et al. (2024) embed semantics in 3DGS for open-vocabulary scene understanding (Yang et al., 2024a). There have also been many works on improving 3D Gaussian itself, such as Yu et al. (2024) improving the reconstruction quality through Mip filter, and Liu et al. (2024) enabling 3D Gaussian to represent super-large scenes through level-of-detail strategy. In terms of practical applications, some work has explored reducing the storage overhead of 3D Gaussian through pruning (Yang et al., 2024b) and entropy coding (Chen et al., 2024).

Although 3D Gaussian is specifically designed for 3D scenes, it has the potential to be a general signal representation method that can be applied to more fields. Zhang et al. (2024) transform the 3D splatting rendering process into a 2D splatting rendering process, building an image representation and compression pipeline based on Gaussian representation. Shin et al. (2024) represent video by foreground Gaussian and background Gaussian, and further achieve video editing. Despite its strong expressive capabilities and fast rendering speed, the exploration of Gaussian in non-3D representation fields is insufficient. How to design better representations to apply 3D Gaussian to different signal like video remains a question.

## 3    METHOD

The implicit assumption of 3DGS for effective scene representation and novel view synthesis is that the dataset describes a 3D scene that can be observed from multiple viewpoints. The assumption may be violated when extend 3DGS to general-purpose representation model (Zhang et al., 2024). To address the issue, we transform the original camera model of 3DGS to TSW orthographic projection (abbreviated as TSW in following sections). TSW is based on the assumption of similarity between neighboring frames, which is inherent characteristics of videos. With TSW, we can easily transform any 3D Gaussian model to video representation model. In section 3.1 we briefly review the background of 3DGS and entropy constrained 3DGS, HAC(Chen et al., 2024). Section 3.2 introduces TSW in details. To enhance the clarity of our method, we illustrate the overall framework of GSVC in Fig. 3. Section 3.3 and section 3.3 explain then time-aware Gaussian generation and optical flow guidance respectively. Section 3.5 summarizes the pipeline of GSVC and introduce the post-compression process.

### 3.1    PRELIMINARIES

3D Gaussian Splatting is designed to synthesis high-quality novel view of a 3D scene efficiently from photos, which is known as inverse rendering. 3DGS models a 3D scene as a large amount of 3D Gaussian points represented by

$$G(\boldsymbol{x}) = \exp(-\frac{1}{2}(\boldsymbol{x} - \boldsymbol{\mu})^{\top}\boldsymbol{\Sigma}^{-1}(\boldsymbol{x} - \boldsymbol{\mu})) \tag{1}$$

where $\boldsymbol{\mu}$ and $\boldsymbol{\Sigma}$ are the mean position and covariance matrix of a 3D Gaussian particle. In rendering process, Gaussian particle are projected to pixel space to obtain 2D Gaussian parameterized by $\boldsymbol{\Sigma}'$

$$\boldsymbol{\Sigma}' = \boldsymbol{J}\boldsymbol{W}\boldsymbol{\Sigma}\boldsymbol{W}^{\top}\boldsymbol{J}^{\top}, \tag{2}$$

where $\boldsymbol{W}$ is view transformation and $\boldsymbol{J}$ is is the Jacobian of the affine approximation of the perspective transformation. Each pixel color $C$ is calculated following $\alpha$-blending

$$C = \sum_{i=1}^{n} \boldsymbol{c}_i \alpha_i \prod_{j=1}^{i-1}(1 - \alpha_j). \tag{3}$$

where $\alpha$ is derived from $\boldsymbol{\Sigma}'$. To maintain the positive semi-definiteness, covariance matrix is further decomposed as rotation $\boldsymbol{R}$ and scaling $\boldsymbol{S}$, i.e. $\boldsymbol{\Sigma} = \boldsymbol{R}\boldsymbol{S}\boldsymbol{S}^{\top}\boldsymbol{R}^{\top}$.

In Scaffold-GS (Lu et al., 2024), all previous parameters are generated from anchor with location $\boldsymbol{x}^a$ and attributes $\mathcal{A} = \{\boldsymbol{f}^a \in \mathbb{R}^{D^a}, \boldsymbol{l} \in \mathbb{R}^6, \boldsymbol{o} \in \mathbb{R}^{3K}\}$, where each component represents anchor feature, scaling and offsets, respectively. HAC (Chen et al., 2024) further exploits mutual relation between $\mathcal{A}$ and $\boldsymbol{f}^h := \text{Interp}(\boldsymbol{x}^a, \mathcal{H})$, to achieve efficient compression, where $\mathcal{H}$ is binary hash grid.

### 3.2    BRIDGING 3D GAUSSIAN SPLATTING AND 2D VIDEO

Few of previous work investigates the method of representing video by 3DGS. Shin et al. (2024) decompose video to foreground Gaussians and background Gaussians. This method requires explicit foreground and background content in the video, but this assumption is not always valid in many videos, such as screen recordings and hand-drawn animations, which are non-natural scene videos. Our core idea is to view 2D videos as a 3D manifold and represent the video using 3D Gaussians directly in the manifold space. This means that every Gaussian point not only affects different pixels at the same time but also affects the same pixel at different times.

Specifically, we view 2D axis xy and time axis t in video as 3D axis xyz in 3D space, and represent the 3D space by Gaussian particles. In render stage, we introduce Toast-like sliding window (TSW) orthographic projection, as shown in Fig. 2c. When rendering frame with time stamp $t_1$, we select all Gaussians with coordinate $z$ falling into $[t_1 - h, t_1 + h]$, which is similar to near clip and far clip in normal orthographic projection. We note space $[t_1 - h, t_1]$ and $[t_1, t_1 + h]$ as $V_f$ and $V_b$ respectively. After culling stage, we use plane $z = t_1$ as camera plane and perform 3D Gaussian

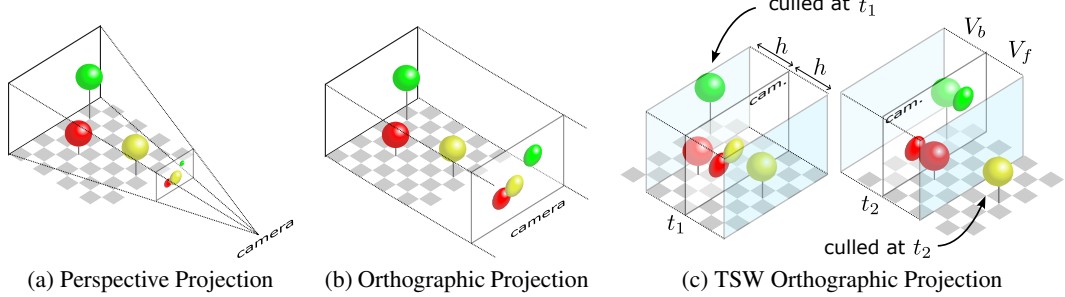

(a) Perspective Projection    (b) Orthographic Projection    (c) TSW Orthographic Projection

Figure 2: Comparison of different camera models.

Splatting rendering process in $V_f$ and $V_b$. Since affine approximation is unnecessary for orthographic projection, the Jacobian matrix $J$ is identity matrix. Eq. 2 can be simplified as

$$\boldsymbol{\Sigma}' = \boldsymbol{W}\boldsymbol{\Sigma}\boldsymbol{W}^\top. \tag{4}$$

To utilizes the bi-directional similarity along time axis, we obtain final reconstructed frame from the average of rendering results of $V_f$ and $V_b$. We show more details and visualization in Appendix A.3.

When using a 3D manifold to represent a 2D video, the redundancy in the t/z axis is often greater than that of the xy axis, which is one of the most important differences from the usual 3D coordinates. The sliding window mechanism improves the representation efficiency by reusing and effectively reduces the redundant information on the time axis. At the same time, this representation method only assumes the similarity in the time axis of the video, and does not require explicit 3D structures, which is more suitable for general video representation scenarios. Moreover, when reconstructing frames, only Gaussian points within the sliding window participate in the rendering process, naturally achieving the goal of stream decoding and random decoding.

### 3.3 TIME-AWARE GAUSSIAN GENERATION

In Scaffold-GS, the generation process of Gaussian points is performed by taking the viewpoint coordinates and anchor point features as inputs, achieving perspective-aware Gaussian Splatting rendering. However, previous studies have shown that directly using coordinates as inputs prevents the network from learning high-frequency features, deteriorates reconstruction quality. Therefore, we use time stamp $t$ of frame and anchor location relative to the camera plane $\delta z^a$ as input after positional encoding (Mildenhall et al., 2020; Tancik et al., 2020)

$$\mathrm{pe}(p) = (\sin(2k\pi p), \cos(2k\pi p))_{k=0}^{L-1}. \tag{5}$$

We set $L = 16$ for both $t$ and $\delta z^a$.

In terms of network architecture, we add feature-wise linear modulation (FiLM) layers (Perez et al., 2018) to MLP. The FiLM layers apply affine transformations to features based on condition inputs. Compared with the original MLP, incorporating FiLM layers increase the network capacity and improve parameter efficiency, which is important to compression task. In our method, the transformation parameters of the FiLM layers are computed from input positional encoding

$$\gamma_{i,c} = g_c(\mathrm{pe}(t), \mathrm{pe}(\delta z^a)), \beta_{i,c} = h_c(\mathrm{pe}(t), \mathrm{pe}(\delta z^a)) \tag{6}$$

where $i$ denotes the index of anchor, $c$ denotes the different FiLM layer of different attributes such as color $\boldsymbol{c}$. The final attribute value are computed from anchor features and FiLM transformation

$$\boldsymbol{f}_i = \mathrm{MLP}_c(f_i^a), \tag{7}$$

$$\hat{\boldsymbol{f}}_i = \mathrm{FiLM}_c(\boldsymbol{f}_i | \gamma_{i,c}, \beta_{i,c}) = \gamma_{i,c}\boldsymbol{f}_i + \beta_{i,c} \tag{8}$$

$$\boldsymbol{c} = \mathrm{MLP}'_c(\hat{\boldsymbol{f}}_i), \tag{9}$$

Here we use color as an example, and the calculation for other attributes are similar.

In addition to the attribute network, we also added a deformation network (Yang et al., 2024c) to enhance the capability of modeling dynamic content. The deformation network takes the positional encoding and anchor feature as input and calculates the dynamic offset delta $\delta o$ of the spawned Gaussians at time $t$.

$$\delta \boldsymbol{o} = \mathrm{MLP}_o(f^a, \mathrm{pe}(t), \mathrm{pe}(\delta z^a)). \tag{10}$$

Given the anchor $\boldsymbol{x}^a$, final position of spawned Gaussians are

$$\{\mu_0, \mu_1, \ldots, \mu_{K-1}\} = \boldsymbol{x}^a + (\delta \boldsymbol{o} + \{o_0, o_1, \ldots, o_{K-1}\})\boldsymbol{l}^a \tag{11}$$

where $\{o_0, o_1, \ldots, o_{K-1}\} \in \mathbb{R}^{K \times 3}$ are the learnable offsets and $\boldsymbol{l}^a$ is the scaling factor associated with that anchor.

### 3.4 Optical Flow guided deformation

High dynamic objects often appear in real-world videos. Currently, research on using 3D Gaussian representation for dynamic content mainly focuses on 4D video (Yang et al., 2024c). Studies on high dynamic content are still insufficient Some work uses optical flow information as auxiliary supervisory signals to guide the training process (Tang et al., 2023). However, such methods usually require an additional network to estimate the optical flow information between reconstructed frames. The performance is affected by the auxiliary network. The explicit representation of 3D Gaussian provides another way to utilize optical flow information.

During the training, we render adjacent two frames simultaneously. For Gaussian points that appear in both frames, we directly supervise their displacement in deformation field using the optical flow information corresponding to the pixel coordinate where the Gaussian point is projected according to

$$\mathcal{L}_{\mathrm{optical},t_1} = \sum_{i=1}^{N} |\hat{\boldsymbol{\mu}}_{i,t_2} - \hat{\boldsymbol{\mu}}_{i,t_1} - u_{\hat{\boldsymbol{\mu}}_{t_1},t_1}|_1, \tag{12}$$

where $\hat{\boldsymbol{\mu}}_{i,t_1}$ $\hat{\boldsymbol{\mu}}_{i,t_2}$ are the coordinates of the $i$-th Gaussian shared between $\{V_f, V_b\}_{t_1}$ and $\{V_f, V_b\}_{t_2}$ at $t_1$ and $t_2$ in pixel space respectively. $u_{\hat{\boldsymbol{\mu}}_{t_1},t_1}$ is the estimated optical flow of corresponding image pixel. The explicit supervision is more effective than implicit information from original image. Another benefit is that the optical flow is estimated independent from training process and can be reused for same video with different model settings.

### 3.5 Training Pipeline, Post training Compression and Entropy Coder

Fig. 3 illustrates the full pipeline of GSVC. We integrated all of the above modules to achieve joint training of reconstruction quality, bit rate, and optical flow constraints.

$$\mathcal{L} = \mathcal{L}_{\mathrm{rec}} + \lambda \frac{1}{N(D^a + 6 + 3K)}(\mathcal{L}_{\mathrm{entropy}} + \mathcal{L}_{\mathrm{hash}}) + \lambda_o \mathcal{L}_{\mathrm{optical}} + \lambda_s \mathcal{L}_{\mathrm{scaffold}} \tag{13}$$

where $\mathcal{L}_{\mathrm{rec}}$ is fusion loss of L1 loss and SSIM loss between ground truth and average render results of $V_f$ and $V_b$. $\mathcal{L}_{\mathrm{optical}}$ is the optical loss. $\mathcal{L}_{\mathrm{scaffold}}$, $\mathcal{L}_{\mathrm{entropy}}$ and $\mathcal{L}_{\mathrm{hash}}$ are similar to previous work. $\lambda$, $\lambda_o$ and $\lambda_s$ are trade-off hyper-parameters among each component.

Despite HAC-based architecture can estimate the end-to-end entropy of most parameters, the anchor coordinates are only quantized during training and not included in the entropy estimation pipeline. The network parameters are also saved in full precision 32-bit. In 3D scene compression, the bit rate of anchor coordinates and network parameters is not high, so saving them in this way will not significantly affect performance. However, in video tasks, we found that the proportion of these two parts has a significant increase. Therefore, we further encoded these parameters to improve the final RD performance.

For the anchor points, we use a geometry-based point cloud compression tool, G-PCC[1] (Schwarz et al., 2018), to ensure the accuracy of the reconstruction. We compress the quantized anchor point coordinates in a lossless manner. Since G-PCC compression reorders the anchor point sequence, we also adjust the order of related attributes before entropy coding them. For all network parameters,

---

[1]https://github.com/MPEGGroup/mpeg-pcc-tmc13

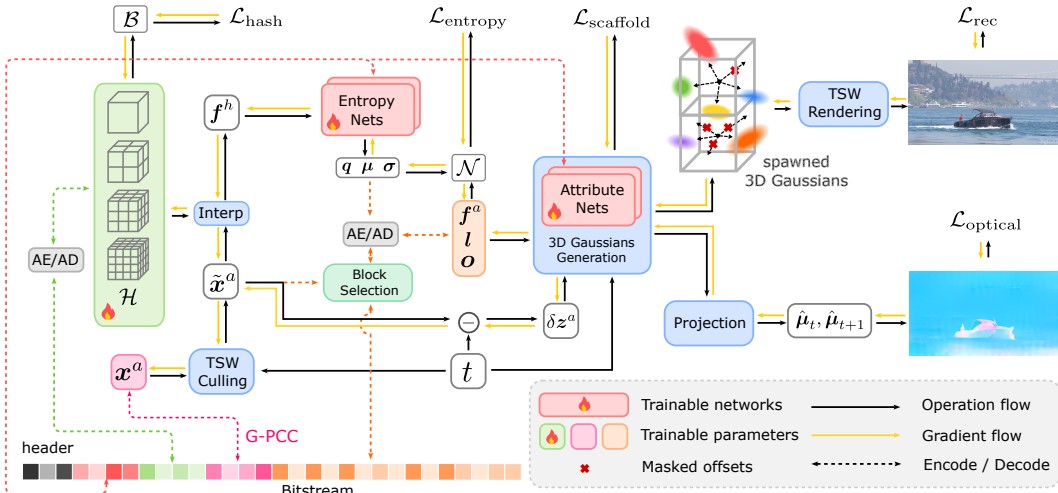

Figure 3: The framework of GSVC. $t$ is the timestamp of frame to be rendered. $q, \mu$ and $\sigma$ are estimated quantization step, mean and scale used in entropy coding. $\mathcal{N}$ denotes we encode anchor attributes $\mathcal{A} = \{f^a \in \mathbb{R}^{D^a}, l \in \mathbb{R}^6, o \in \mathbb{R}^{3K}\}$ by Normal distribution. Similarly, $\mathcal{B}$ represent Bernoulli, used to encode binary hash grid $\mathcal{H}$. AE/AD represent ANS (Duda et al., 2015) encoder and decoder. Note the offsets mask is also decoded from bitstream.

following previous work (Chen et al., 2021), we directly quantize them to 8 bits. After this post-processing, the compression ratio can be further improved.

Furthermore, we implemented a GPU-based entropy codec according to ANS theory (Duda et al., 2015; Bamler, 2022). Most of entropy coders utilize the powerful serial computation ability of CPU cores but misalign with the requirements of emerging compressor building on neural network, which conduct most of the computation on GPU. Frequent communication between CPU and GPU will deteriorate the coding speed, let alone the bottleneck of pure serial processing. In proposed entropy codec, we take advantage of parallelization by probing decoding. Appendix A.1 explains the detail designs.

## 4 EXPERIMENTS

### 4.1 EXPERIMENTS SETUP

**Dataset** All experiments are performed on popular UVG dataset (Mercat et al., 2020). We select first 7 videos as dataset, consisting videos with resolution of $1920 \times 1080$ and $600 \times 6 + 300 = 3900$ frames in total, following previous works (Chen et al., 2021; Lee et al., 2023). In GSVC, we use optical flow estimated by pre-trained model of VideoFlow (Shi et al., 2023).

**Metrics** We use the peak signal-to-noise ratio (PSNR) in RGB 4:4:4 and MS-SSIM (Wang et al., 2003) as distortion quality metric, which are the most widely used metric in compression. We also report LPIPS (Zhang et al., 2018) in main experiment, which is a popular perceptual metric to measure realism. Bit-per-pixel (BPP) is used as coding efficiency metric.

**Implementation details** We implement differentiable rasterizer with orthographic projection based the repository of HAC. The sliding window mechanism and GSVC are implemented with PyTorch. In experiments of 3DGS family methods, we build Structure-from-Motion (SfM) cloud directly from UVG dataset by COLMAP (Schönberger & Frahm, 2016; Schönberger et al., 2016). For video fails to converge or generates multiple SfM clouds, we skip the video in experiments. For TSW variant of 3DGS family methods and GSVC, there is no trivial method to obtain a meaningful initialization like SfM. We simply start training from random initialized point cloud. We assign camera position of each frame as $(0, 0, t_n)$, where $t_n$ is normalized timestamp of corresponding frame. We set $h = 0.4$ for HoneyBee, $h = 0.2$ for ShakeNDry and $h = 0.05$ for other videos. We reproduce the results of

Table 1: Numerical results of all methods. For vanilla 3DGS (Kerbl et al., 2023), Scaffold-GS (abbreviated as S-GS) (Lu et al., 2024) and HAC (Chen et al., 2024), COLMAP fails to reconstruct valid scene on HoneyBee, Jockey and ShakeNDry so we skip these videos. Subscript TSW denotes the TSW variants of these methods. The best and second best results are highlighted in red and yellow cells. The size is measured in MB.

| Data | Metrics | 3DGS | S-GS | HAC | 3DGS$_{TSW}$ | S-GS$_{TSW}$ | HAC$_{TSW}$ | GSVC |
|------|---------|------|------|-----|--------------|--------------|-------------|------|
| Beauty | PSNR ↑ | 25.87 | 28.52 | 26.27 | 28.54 | 29.21 | 29.77 | 30.25 |
| | SSIM ↑ | 0.8102 | 0.8367 | 0.8116 | 0.8414 | 0.8426 | 0.8549 | 0.8532 |
| | LPIPS ↓ | 0.6033 | 0.6034 | 0.6191 | 0.6022 | 0.6126 | 0.5925 | 0.5745 |
| | SIZE ↓ | 28.10 | 4.930 | 1.493 | 25.18 | 5.995 | 2.316 | 1.107 |
| | FPS ↑ | 126.3 | 125.3 | 154.6 | 93.05 | 84.32 | 89.04 | 83.37 |
| Bospho. | PSNR ↑ | 23.47 | 26.59 | 25.42 | 29.03 | 29.04 | 32.00 | 33.19 |
| | SSIM ↑ | 0.7461 | 0.8815 | 0.8542 | 0.8726 | 0.8753 | 0.9260 | 0.9517 |
| | LPIPS ↓ | 0.4427 | 0.2601 | 0.2985 | 0.3503 | 0.3551 | 0.2660 | 0.1780 |
| | SIZE ↓ | 121.0 | 7.990 | 2.105 | 24.80 | 5.903 | 5.091 | 2.050 |
| | FPS ↑ | 53.24 | 92.61 | 137.0 | 77.63 | 60.81 | 50.82 | 67.03 |
| Honey. | PSNR ↑ | - | - | - | 35.23 | 31.00 | 36.50 | 37.90 |
| | SSIM ↑ | - | - | - | 0.9755 | 0.9525 | 0.9801 | 0.9830 |
| | LPIPS ↓ | - | - | - | 0.1844 | 0.2520 | 0.1728 | 0.1571 |
| | SIZE ↓ | - | - | - | 23.87 | 5.903 | 3.169 | 2.313 |
| | FPS ↑ | - | - | - | 81.33 | 98.48 | 76.73 | 39.63 |
| Jockey | PSNR ↑ | - | - | - | 21.71 | 22.93 | 22.76 | 29.38 |
| | SSIM ↑ | - | - | - | 0.7399 | 0.7673 | 0.7760 | 0.8956 |
| | LPIPS ↓ | - | - | - | 0.5342 | 0.5298 | 0.4975 | 0.3777 |
| | SIZE ↓ | - | - | - | 22.48 | 6.741 | 3.062 | 2.429 |
| | FPS ↑ | - | - | - | 83.96 | 49.17 | 41.42 | 61.70 |
| Ready. | PSNR ↑ | 21.30 | 23.51 | 22.43 | 18.27 | 18.88 | 19.75 | 26.96 |
| | SSIM ↑ | 0.8044 | 0.8655 | 0.8354 | 0.6143 | 0.6349 | 0.6943 | 0.9263 |
| | LPIPS ↓ | 0.3120 | 0.2718 | 0.2890 | 0.6213 | 0.6432 | 0.5381 | 0.2326 |
| | SIZE ↓ | 631.4 | 85.3900 | 21.63 | 26.45 | 7.047 | 6.433 | 6.350 |
| | FPS ↑ | 76.21 | 98.82 | 119.3 | 71.38 | 76.62 | 51.45 | 51.90 |
| Shake. | PSNR ↑ | - | - | - | 29.75 | 30.41 | 31.75 | 34.22 |
| | SSIM ↑ | - | - | - | 0.9068 | 0.9113 | 0.9272 | 0.9470 |
| | LPIPS ↓ | - | - | - | 0.3279 | 0.3218 | 0.2719 | 0.2239 |
| | SIZE ↓ | - | - | - | 24.16 | 9.824 | 3.729 | 3.153 |
| | FPS ↑ | - | - | - | 73.39 | 83.22 | 77.81 | 42.93 |
| Yacht. | PSNR ↑ | 20.31 | 23.37 | 20.59 | 23.84 | 23.94 | 25.69 | 28.61 |
| | SSIM ↑ | 0.6511 | 0.7705 | 0.6904 | 0.7779 | 0.7851 | 0.8491 | 0.9201 |
| | LPIPS ↓ | 0.5379 | 0.4239 | 0.4855 | 0.4610 | 0.4703 | 0.3684 | 0.2469 |
| | SIZE ↓ | 50.77 | 38.60 | 5.485 | 24.34 | 5.902 | 5.502 | 3.7015 |
| | FPS ↑ | 47.35 | 83.72 | 123.2 | 68.13 | 66.4746 | 50.3142 | 64.00 |

NeRV (Chen et al., 2021) according to original hyper-parameters setting but in per video way. We conduct all experiments on a single RTX 3090. More details can be found in Appendix A.2.

## 4.2 QUANTITY AND QUALITY RESULTS

Table 1 demonstrates the result among 3DGS family methods and proposed method. Not surprisingly, vanilla 3DGS, Scaffold-GS and HAC cannot be considered a valid video representation model, failing to reconstruct all videos in dataset. It is obvious that not all videos contain a 3D scene, which is a key modeling assumption of 3DGS. For TSW variants of these methods, we only require adjacent similarity among frames, which effectively transform these methods to video representation models. HAC$_{TSW}$ achieves second best results in many metrics.

But we also notice that for ReadySetGo, TSW variants fail to surpass vanilla methods. The result indicates the insufficiency of representing video by TSW only for videos including high dynamic

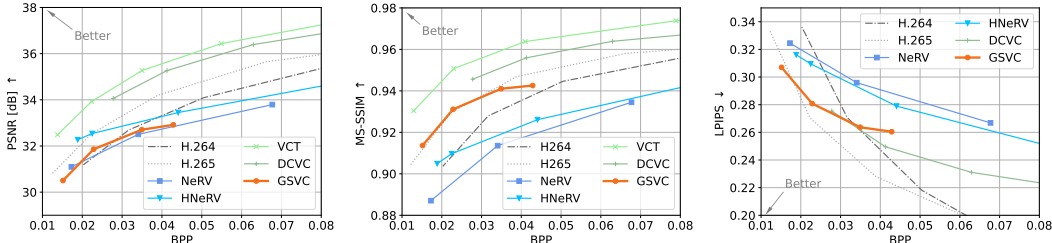

Figure 4: Comparison with other methods on UVG dataset. GSVC significantly outperforms NeRV (Chen et al., 2021) and HNeRV (Chen et al., 2023) on MS-SSIM and LPIPS and achieves comparable MS-SSIM and LPIPS performance with H.265 (*veryslow*) at low bit rate region.

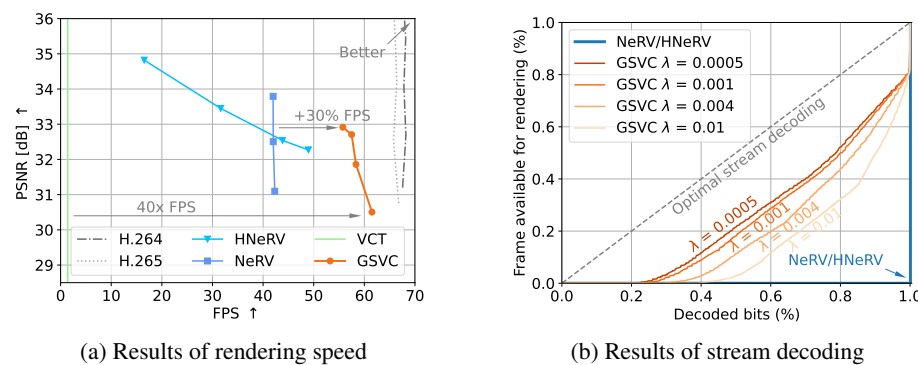

(a) Results of rendering speed

(b) Results of stream decoding

Figure 5: Comparison with other methods on practical metrics. GSVC outperforms all learning-based methods on rendering speed. We omit DCVC (Li et al., 2021) here due to its slow autoregressive decoding ($\ll 1$ FPS). In stream decoding, when bit stream is completely decoded, all frames are available for rendering in all methods.

contents. Cooperating with proposed time-aware generation and optical guided deformation, GSVC achieves best result in majority of metrics except for FPS. Since GSVC requires rendering twice for each frame, this result is reasonable.

Fig. 4 demonstrates the comparison of GSVC and other methods including H.264, H.265, DCVC (Li et al., 2021), VCT (Mentzer et al., 2022), NeRV (Chen et al., 2021) and HNeRV (Chen et al., 2023). We use $\lambda = \{0.1, 0.04, 0.001, 0.005\}$ and averages the results over same $\lambda$. GSCV attains comparable performance in PSNR and significantly better performance in MS-SSIM and LPIPS than NeRV and HNeRV. This is a positive proof-of-concept of applying 3DGS in video compression. We present more results and visualizations of all methods in Appendix A.5.

## 4.3 RENDERING SPEED AND STREAM DECODING

Rendering speed is important for real-world application. Note here we refer decoding as the process of entropy decoding for GSVC and Huffman decoding for NeRV. Rendering speed only account for reconstruction since the decoding process can be easily overlapped by multiprocessing. Fig. 5a shows GSVC achieves 30 % to 40% higher FPS than NeRV and over $40\times$ FPS than VCT, demonstrating the efficiency of our method.

Stream decoding is another important feature for practical usage. Fig. 5b shows the stream decoding result. NeRV must decoding whole network before rendering frames. But for GSVC, after receiving common information including meta, anchor position and weights of network, the remaining bitstream can be decoded on demand. Besides, we also notice that better stream decoding performance is achieved when using high bit rate setting (smaller $\lambda$). This is because the size of common information is similar at different bitrates.

Table 2: Ablation results. We report BD-rate (%) (Bjøntegaard, 2001) of all settings relative to GSVC. Negative value denote the setting has better RD performance than GSVC. Failing to calculate BD-rate means the highest PSNR among all $\lambda$ of the setting is smaller than the lowest PSNR of GSVC. w/o deformation means removing deformation network. w/o FiLM represents using original MLP instead of network with FiLM layer.

| Settings | Beauty | Bospho. | Honey. | Jockey | Ready. | Shake. | Yacht. |
|---|---|---|---|---|---|---|---|
| w/o optical guidance | **-5.374** | 2.618 | **-5.060** | 81.21 | 78.37 | **-10.56** | 1.753 |
| w/o post compression | 72.81 | 42.75 | 68.66 | 40.68 | 32.24 | 58.49 | 39.31 |
| w/o deformation | 0.347 | - | 38.15 | - | - | 41.39 | - |
| w/o FiLM | 64.82 | 134.1 | 3.039 | - | - | 64.67 | - |

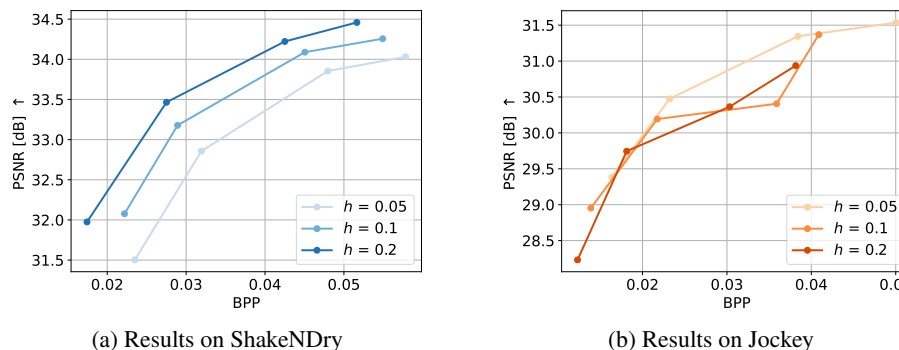

(a) Results on ShakeNDry          (b) Results on Jockey

Figure 6: Ablation results of $h$. ShakeNDry contains an almost static background while Jocky is a highly dynamic video.

## 4.4 ABLATION STUDY

Table. 3 demonstrates the effectiveness of different components of GSVC. In general, all the proposed modules effectively improve the final performance except optical flow guidance. For videos with high dynamic contest, optical flow guidance is critical. For the data with negative effects, further inspection reveals that they are mainly caused by noise in the optical flow estimation results. This also indirectly indicates the importance of optical flow information. How to overcome these inevitable noises is a direction worth exploring in the future.

Fig. 6 shows the impact of the $h$ in TSW. For video that is close to static, a larger $h$ performs better, while for dynamic video, a smaller $h$ performs better. Fig. 6b also indicates that large $h$ may lead to unstable training for dynamic video. Due to the non-uniform dynamic characteristics of videos, manually specifying $h$ or determine $h$ by warm-up training for all frames is far from optimal. How to adaptively assign $h$ to each frame according to dynamics of adjacent frames is another valuable question.

## 5 DISCUSSION

**Conclusion** In this paper, we propose a novel TSW orthographic projection to effectively transform any 3D Gaussian model to a video representation model. Based on TSW we further introduce a practical video compressor GSVC. As a proof-of-concept, we explore the characteristics of 3DGS-based video codec by extensive experiments. Our approach has demonstrated promising RD performance with practical fast rendering and stream decoding. With the advancements of 3DGS compression techniques (Wang et al., 2024), 3DGS-based video codec is worth exploring further in the future.

**Limitation** Similar to other INR-based methods, GSVC still relies on a per-video training encoding process, which limits its application in real-time encoding tasks. Moreover, the proposed model still has room for improvement. These limitations also suggest more directions for further exploration.

ACKNOWLEDGMENTS

This work is supported in part by the National Natural Science Foundation of China under grant 62171248, 62301189, the project of Peng Cheng Laboratory (PCL2023A08), and Shenzhen Science and Technology Program under Grant KJZD20240903103702004, JCYJ20220818101012025, RCBS20221008093124061, GXWD20220811172936001.

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

# A    APPENDIX

## A.1    GPU-BASED ANS CODEC

The ANS codec are often considered to have higher encoding and decoding speeds while keeping similar coding efficiency to arithmetic coding. We implement our codec by extending the algorithm introduced by Bamler (2022) to massively parallel environment. Algo. 1 and Algo. 2 demonstrates the encoding and decoding process.

---

**Algorithm 1** Encoding process

1: **procedure** ENCODE_SYMBOLS($s, \mu, \sigma, N$)
2:     $p_s \leftarrow \text{PMF}(s, \mu, \sigma)$
3:     $c_s \leftarrow \text{CDF}(s, \mu, \sigma)$
4:     $head \leftarrow 0$                                        ▷ $head$ is an unsigned 32 bit integer
5:     $bitstream \leftarrow []$                                ▷ Empty array to store bit stream
6:     $mask \leftarrow 2^{16} - 1$
7:     $cursor \leftarrow 0$
8:     **for** $i \leftarrow 0, N - 1$ **do**
9:         $p \leftarrow p_s[i]$
10:        $c \leftarrow c_s[i]$
11:       **if** $(head >> 16) > p$ **then**
12:           $cursor \leftarrow cursor + 1$
13:           $bitstream[cursor] \leftarrow head \& mask$
14:           $head \leftarrow head >> 16$
15:       **end if**
16:       $z \leftarrow head \mod p + c$
17:       $head \leftarrow \lfloor head/p \rfloor;$
18:       $head \leftarrow head << 16 | z$
19:     **end for**
20:     **while** $head > 0$ **do**                  ▷ Transfer the reaming bits in $head$ to $bitstream$
21:       $cursor \leftarrow cursor + 1$
22:       $bitstream[cursor] \leftarrow head \& mask$
23:       $head \leftarrow head >> 16$
24:     **end while**
25:     **return** $bitstream$
26: **end procedure**

---

In Algo. 1, $s$ is symbols vector to be encoded with $N$ elements. $\mu, \sigma$ are distribution parameters associated with each symbol. On GPU, we can quickly compute probability mass function (PMF) value and cumulative density function (CDF) value of each symbol, shown in line 1 to 2 in the algorithm. The remain part of algorithm follows the common ANS encoding process in serial fashion.

Algo. 2 demonstrates the decoding process of our codec. Different from common ANS decoder, we suppose all symbols in support set are valid symbol to be decoded. Obviously only the trial holds the correct symbol will succeed. For massively parallel GPU, performing all attempts will not take much longer time than completing just one. In serial decoding, it is difficult to decode the correct symbol with just a few trials. Therefore, even if the GPU thread is slower than CPU thread, our method can still achieve faster decoding speed, especially with larger support sets.

In Algo. 2 PMF() and CDF() compute the full PMF and CDF for whole support set of all symbols to be decoded. Fig. 7 demonstrates the encoding and decoding speed of proposed codec and torchac (Mentzer et al., 2019) on random generated message with 20000 symbols. Since our method does not compute full PMF values and CDF values, the acceleration is more significant in encoding stage.

---

**Algorithm 2** Parallel decoding process

---

1: **procedure** DECODE_SYMBOLS($bitstream$,cursor, $\boldsymbol{\mu}, \boldsymbol{\sigma}, N$)
2:     $\boldsymbol{P_s} \leftarrow \text{PMF}(\boldsymbol{\mu}, \boldsymbol{\sigma})$                                  ▷ $\boldsymbol{P_s}, \boldsymbol{C_s}$ are computed outside the function,
3:     $\boldsymbol{C_s} \leftarrow \text{CDF}(\boldsymbol{\mu}, \boldsymbol{\sigma})$                                          ▷ we put them here for demonstration.
4:     **while** $cursor \geq 0$ **and** $head >> 16 == 0$ **do**
5:         $token \leftarrow bitstream[cursor]$
6:         $cursor \leftarrow cursor - 1$
7:         $head \leftarrow head << 16 | token$
8:     **end while**
9:     \_\_shared\_\_ $symbols \leftarrow []$                            ▷ Empty array to store decoded symbols
10:     \_\_shared\_\_ $s$                                     ▷ Current decoded symbol
11:     \_\_shared\_\_ $r$                                       ▷ Residue in decoding
12:     $mask \leftarrow 2^{16} - 1$
13:     **for** $i \leftarrow 0, N - 1$ **do**
14:         $\boldsymbol{p} \leftarrow \boldsymbol{P_s}[i,:]$
15:         $\boldsymbol{c} \leftarrow \boldsymbol{C_s}[i,:]$
16:         $z \leftarrow head \& mask$
17:         $head \leftarrow head >> 16$
18:         $tid \leftarrow$ thread ID of CUDA kernel
19:         $lb = \boldsymbol{c}[tid]$
20:         $ub = \boldsymbol{c}[tid + 1]$
21:         **if** $z >= lb$ **and** $z < ub$ **then**             ▷ Only thread holds correct symbol pass the check
22:             $s \leftarrow tid$
23:             $r \leftarrow z - lb$
24:             $symbols[i] = s$
25:         **end if**
26:         block.sync()                                    ▷ Sync all threads of CUDA kernel
27:         $head \leftarrow head \times \boldsymbol{p}[s] + r$
28:         **if** $(head >> 16) == 0$ **and** $cursor \geq 0$ **then**
29:             $token \leftarrow bitstream[cursor]$
30:             $cursor \leftarrow cursor - 1$
31:             $head \leftarrow head << 16 | token$
32:         **end if**
33:     **end for**
34:     **return** $symbols$
35: **end procedure**

---

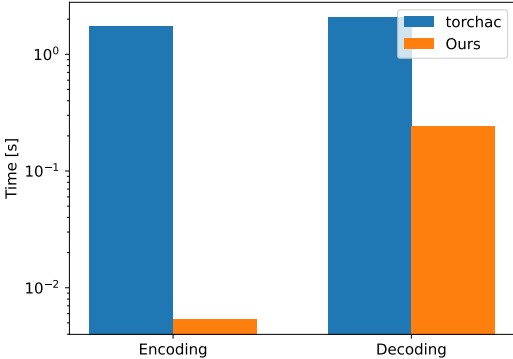

Figure 7: Comparison results. Our method significantly faster than torchac

### A.2 More implementation details

### A.2.1 Hyper-parameters settings

For original 3DGS, Scaffold-GS and HAC, we use the default setting. Note $\lambda_e$ is a tunable hyper-parameter to control the rate-distortion trade-off. In order to obtain more intuitive results, we set $\lambda_e = 10^{-9}$ to Beauty, $\lambda_e = 10^{-8}$ to Bosphorus and $\lambda_e = 10^{-5}$ to the others.

In experiments of TSW variants, we sample coordinates used to initialize Gaussian from uniform distribution. We set initial anchor number to 100000 for GSVC and 20000 to the others. Note GSVC achieves lower bit rate with more initialization anchors. For HAC$_{\text{TSW}}$, we set $\lambda_e = 0.001$ for all videos.

Besides, we set densification threshold to 0.0005, which limited the densification speed in some high dynamic videos. We also set grid dimension to 8 to increase the capacity of hash grid. For the remaining hyper-parameters of GSVC shared with HAC, including learning rates and their scheduler, we follow the original setting.

### A.2.2 Training schedule

In all experiments, we train GSVC 40000 iterations in total. In first 10000 iterations, we use full precision training and adding uniform noise to simulate quantization error from iteration 10000 to 35000. After 35000 iterations, we employ a straight-through estimator (Theis et al., 2022) to align with real quantization. The entropy constraints is added from iteration 15000. We apply anchor densification and pruning between iteration 1500 and 25000. We also pause the operations at the beginning of quantization training i.e. iterations between 10000 and 11000 to improve training stability.

### A.2.3 building SfM clouds from video

Because SfM clouds initialization is necessary for 3DGS family methods, we generate the point clouds using COLMAP[2] (Schönberger & Frahm, 2016; Schönberger et al., 2016).

```
import pycolmap
import pathlib

output_path = pathlib.Path('/path/to/ouput_dir')
image_dir = pathlib.Path('/path/to/frames_dir')
database_path = output_path / 'database.db'
pycolmap.extract_features(database_path, image_dir)
pycolmap.match_exhaustive(database_path)
maps = pycolmap.incremental_mapping(
    database_path, image_dir, output_path
)
maps[0].write(output_path)
```

Listing 1: Snippet used to build SfM clouds from video frames

We successfully build SfM for Bosphorus and YachtRide. COLMAP extract multiple SfMs from Beauty, Jockey and ReaySetGo. Since the biggest SfM of Beauty and ReadySetGo includes more than 500 frames of original video, we consider them as valid SfM. For HoneyBee and ShakeNDry, COLMAP fails to converge.

### A.2.4 Details of baseline methods

For H.265 and H.264 we follow the settings in DCVC(Li et al., 2021) and average the results over same *qp* value. For DCVC we use the pre-trained models to evaluate the performances. Instead of

---

[2]https://github.com/colmap/colmap

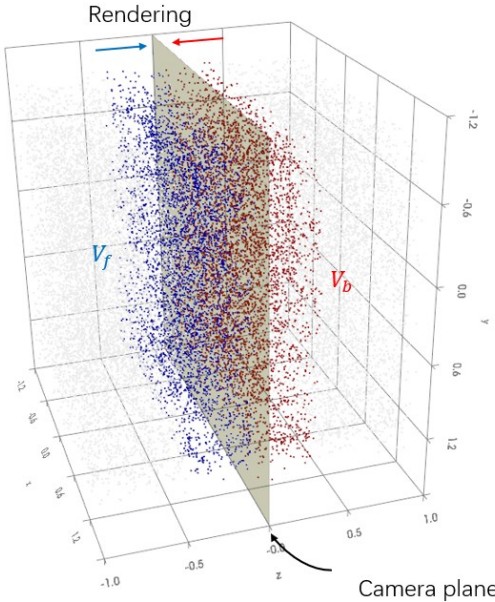

Figure 8: Example of rendering one frame in real anchors cloud.

testing only 120 frames in original paper, we test all frames in UVG dataset. The results of VCT (Mentzer et al., 2022) is copied from original paper. Different from other methods using models training with same specified loss function in all experiments, the results of VCT on PSNR and MS-SSIM are from corresponding models *i.e.* test PSNR using models trained for PSNR and test MS-SSIM using models trained for MS-SSIM.

### A.3 More Details about Rendering

Fig. 8 illustrates a real example of rendering one frame. To render a frame at time $t_0$, we use camera plane $z = t_0$. Based on the camera plane, we cull anchors that do not participate in the rendering (gray points in Fig. 8). For anchors fall into $V_f$ and $V_b$, we render them separately. The rendering follows standard 3D Gaussian Splatting rendering procedure, except skipping view frustum culling and replacing perspective projection matrix with orthographic projection matrix (Eq. 2).

Fig. 9 demonstrates progressive rendering results. In each setting, we use a subset of Gaussians in $V_f$ and $V_b$ to render a frame, according to Gaussian particles' distance to camera plane. For example, when $d = 0.03$, we only select Gaussians with distance smaller than 0.03. It is obvious that Gaussians near camera plane tend to focus on dynamic objects *i.e.* dog in the scene, while distant Gaussians pay more attention to static background, which is shared by more frames.

### A.4 Results of Encoding Time

Table 3 demonstrates the comparison of encoding time. GSVC outperforms other INR-based methods. It should be noted that smaller total rendered frames during training does not necessarily mean faster converging. Training time depends on model size, model structure, and whether entropy constraint is enforced during training. For GSVC, the time of one iteration with entropy constraint is longer than without entropy constraint due to the consumption of the entropy network. Since there is no accepted metric like FLOPS to measure the computational complexity of 3D Gaussian Splatting rendering, we use GPU utilization from nvidia-smi as a reference. A utilization below 100% indicates that GSVC's computational consumption is lower than other methods. However, it also suggests that the GPU's capability is not fully tapped. This limitation calls for further exploration and improvement in the future.

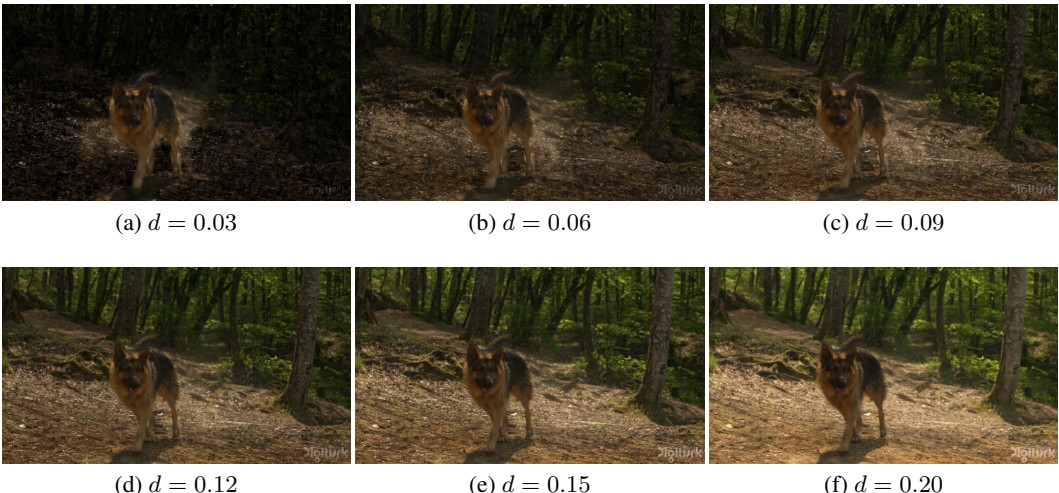

(a) $d = 0.03$        (b) $d = 0.06$        (c) $d = 0.09$

(d) $d = 0.12$        (e) $d = 0.15$        (f) $d = 0.20$

Figure 9: Progressive rendering results. $d$ is the max distance between camera plane and Gaussian particles participating in rendering.

Table 3: Training/Encoding time comparison. Frames represents number of total frames rendered during training. We show the results of a typical 600 frames video in UVG dataset. Training time of HNeRV is sensitive to model size *i.e.* bpp, so we demonstrate a range here. GPU Util. is from nvidia-smi.

| Method | Frames | Time | GPU Util. |
|---|---|---|---|
| NeRV | 240k | 22ks (6h6m) | 100% |
| HNeRV | 180k | 13.8ks~28.2ks (3h50m~7h50m) | 100% |
| GSVC | **80k** | **12.4ks (3h26m)** | 85% |

## A.5 VISUALIZATION

We demonstrate the visualization results in following pages.

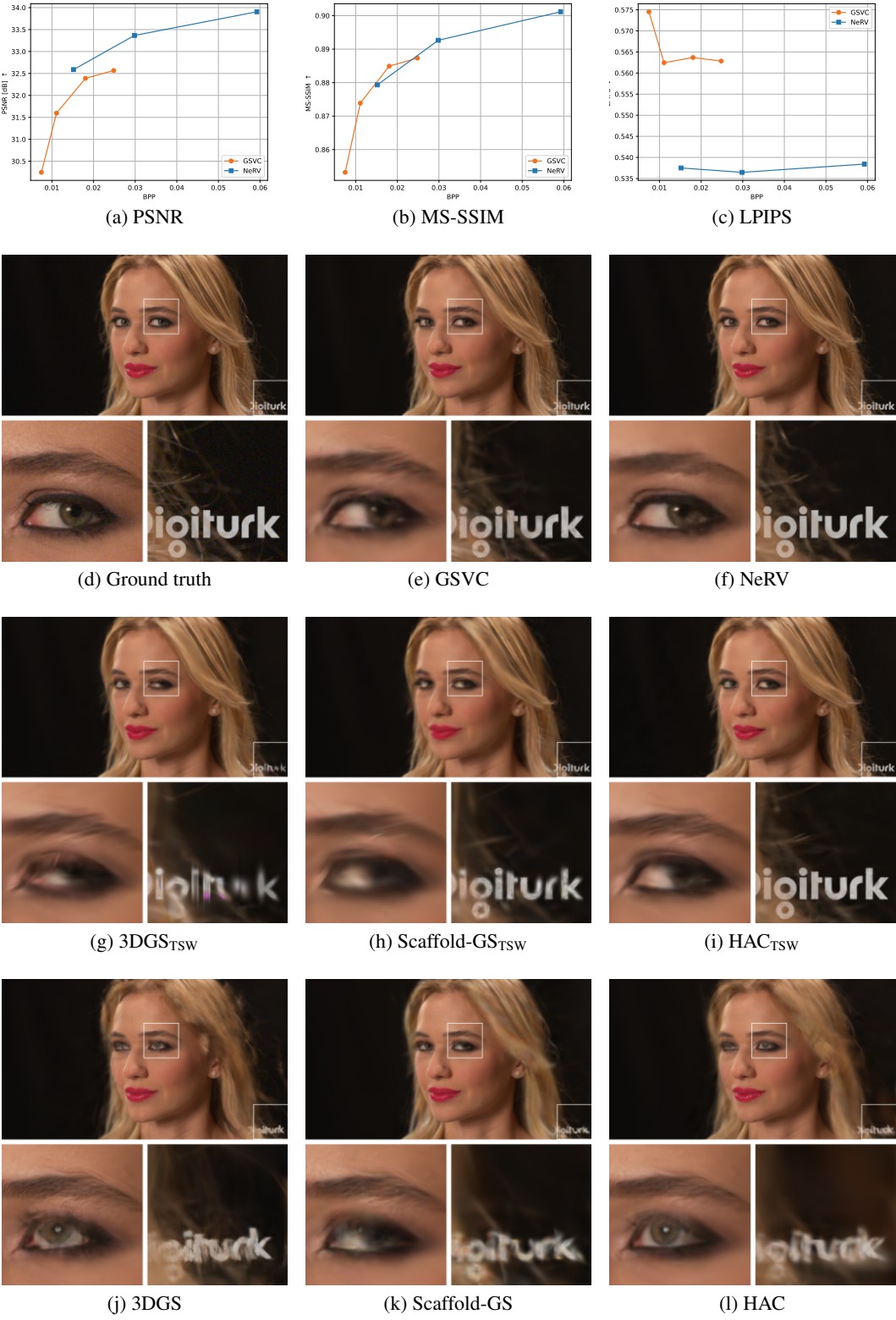

Figure 10: Beauty

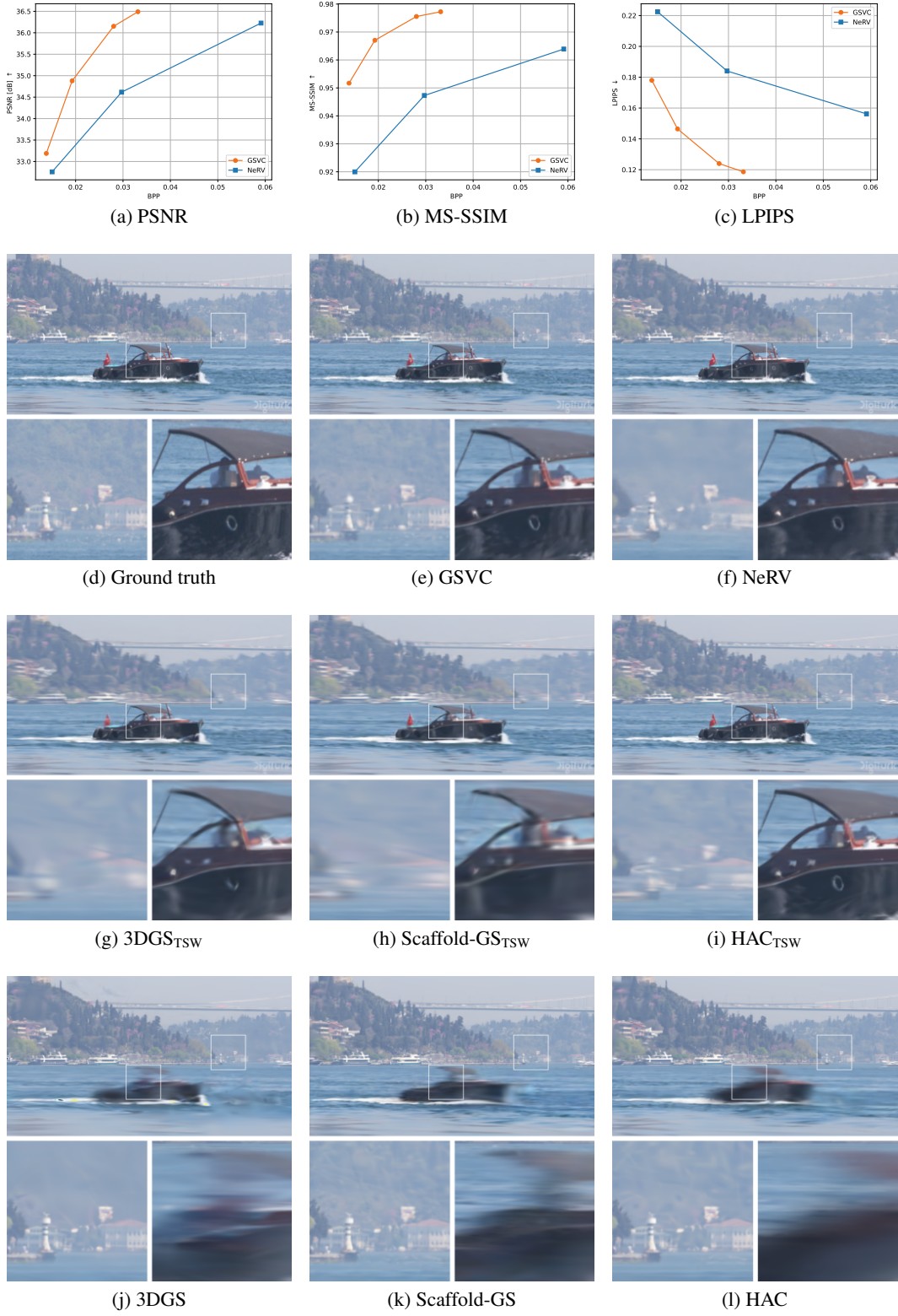

Figure 11: Bosphorus

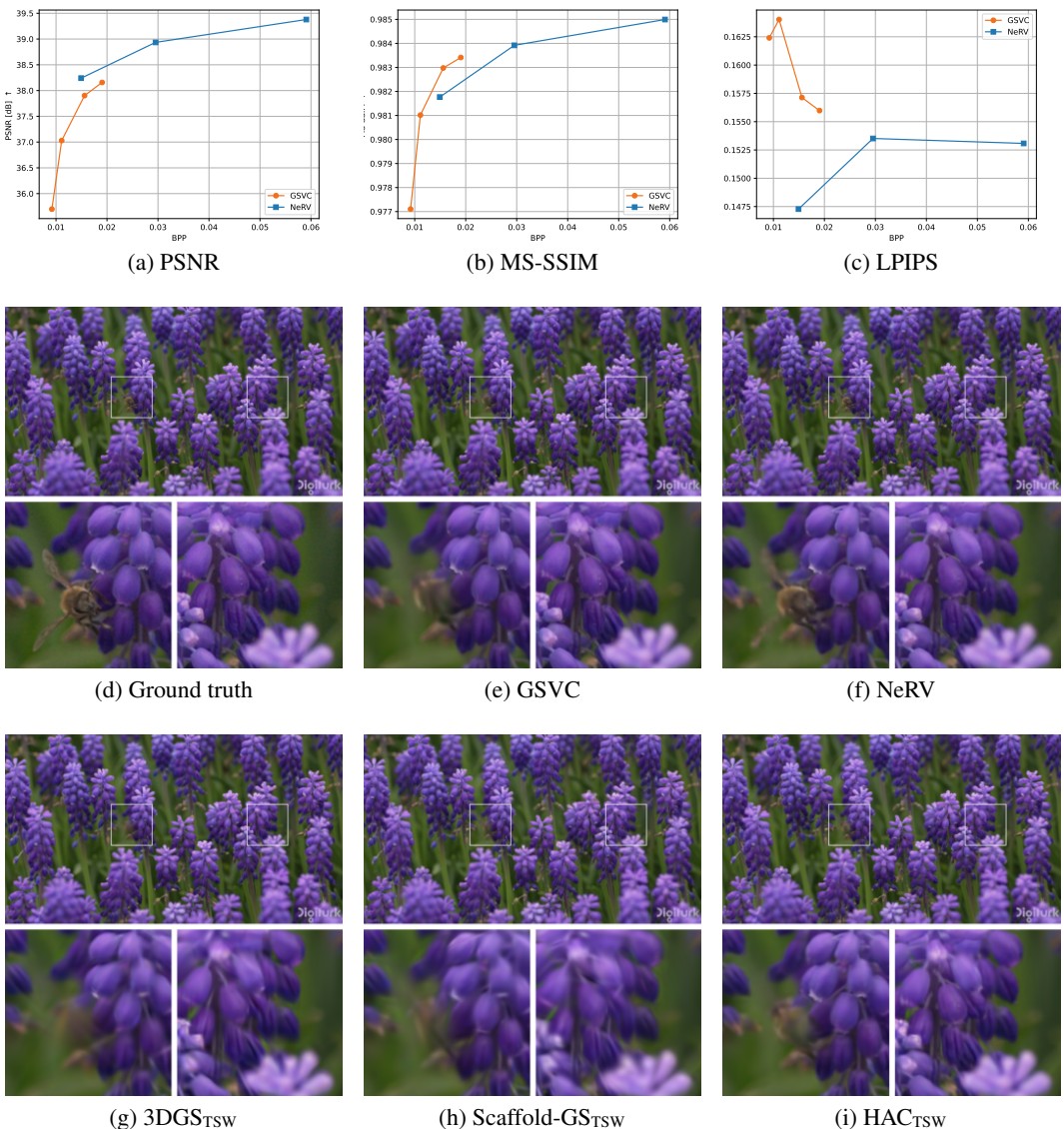

Figure 12: HoneyBee

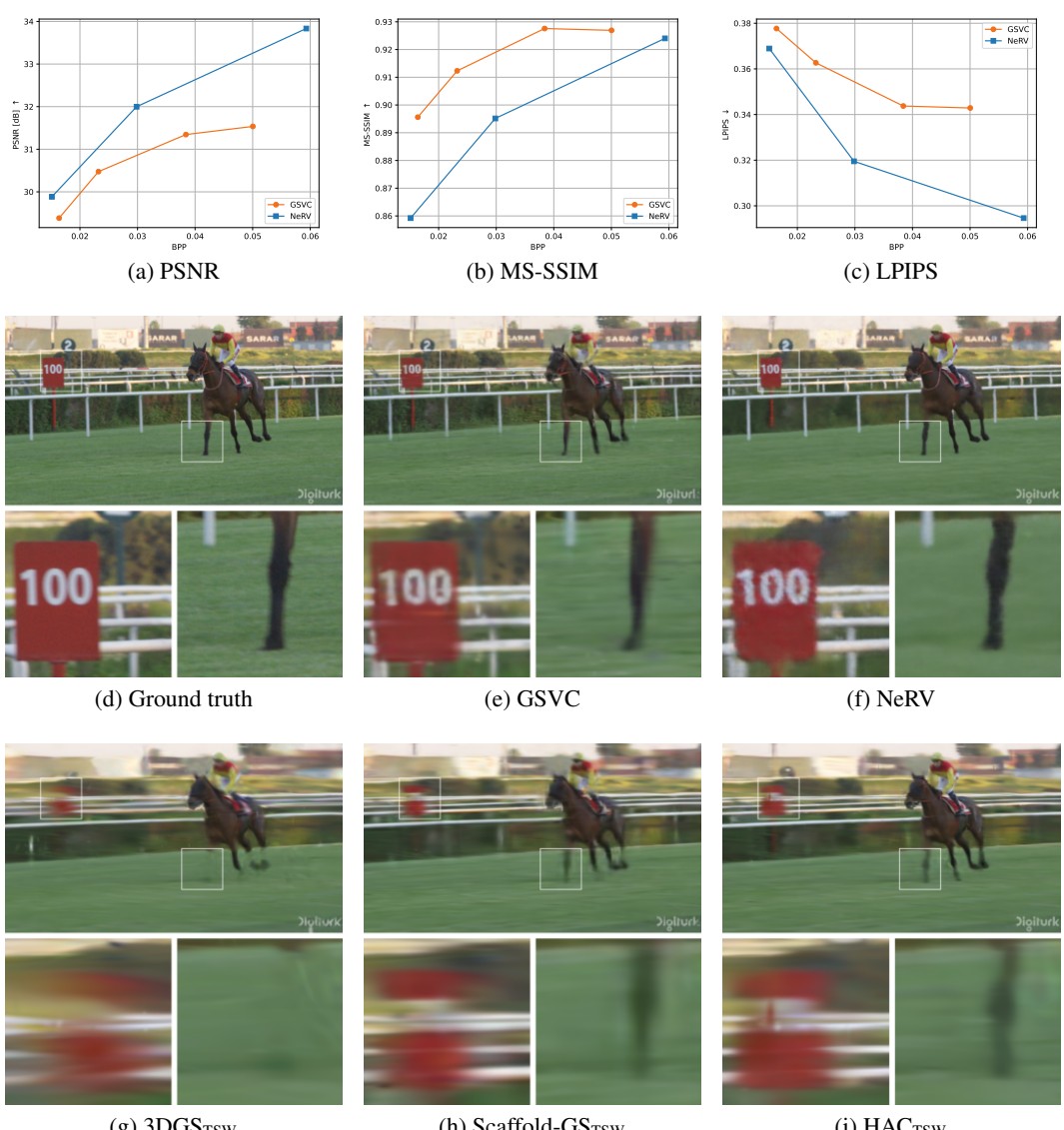

(a) PSNR

(b) MS-SSIM

(c) LPIPS

(d) Ground truth

(e) GSVC

(f) NeRV

(g) 3DGS$_{TSW}$

(h) Scaffold-GS$_{TSW}$

(i) HAC$_{TSW}$

Figure 13: Jockey

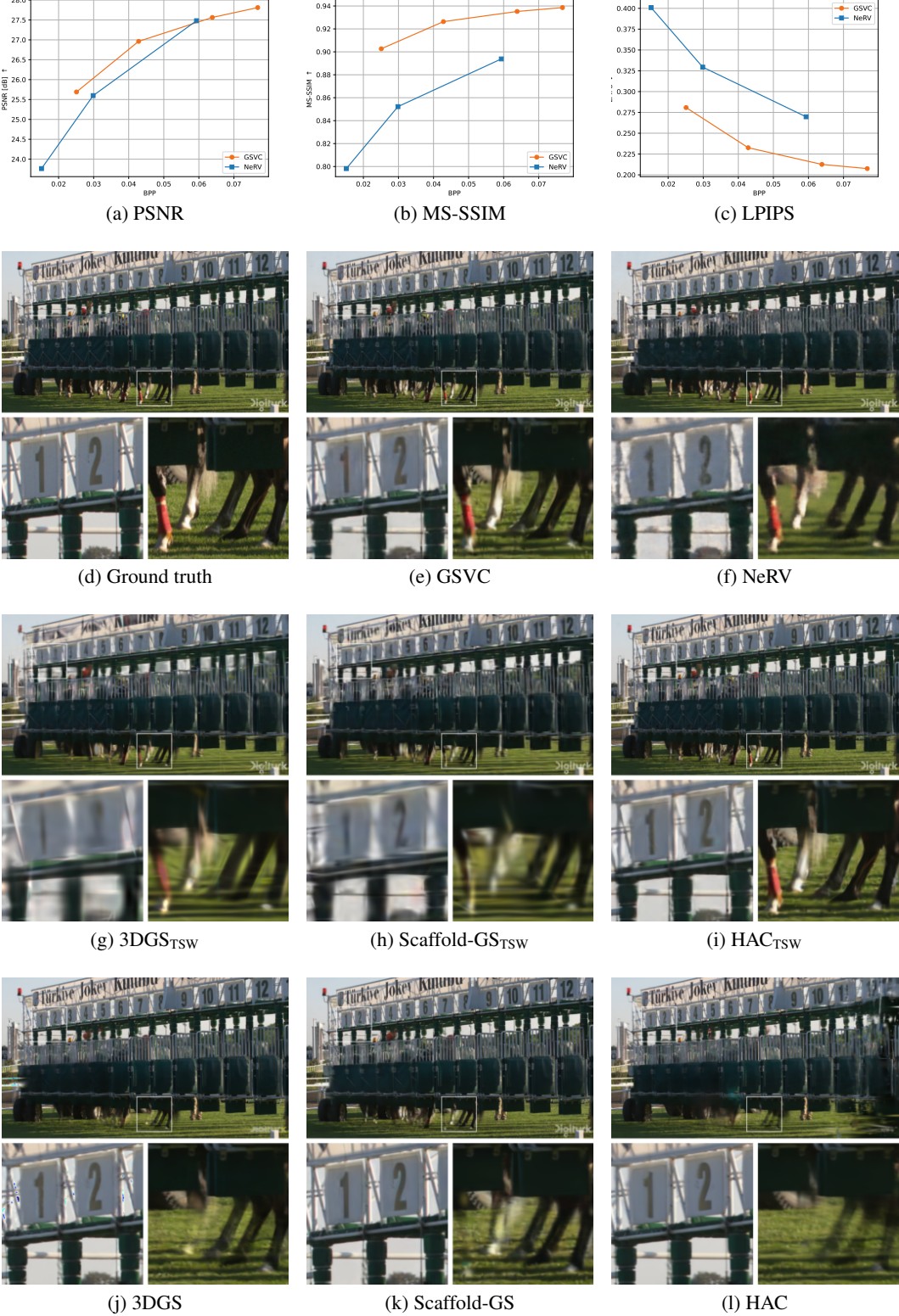

Figure 14: ReadySetGo

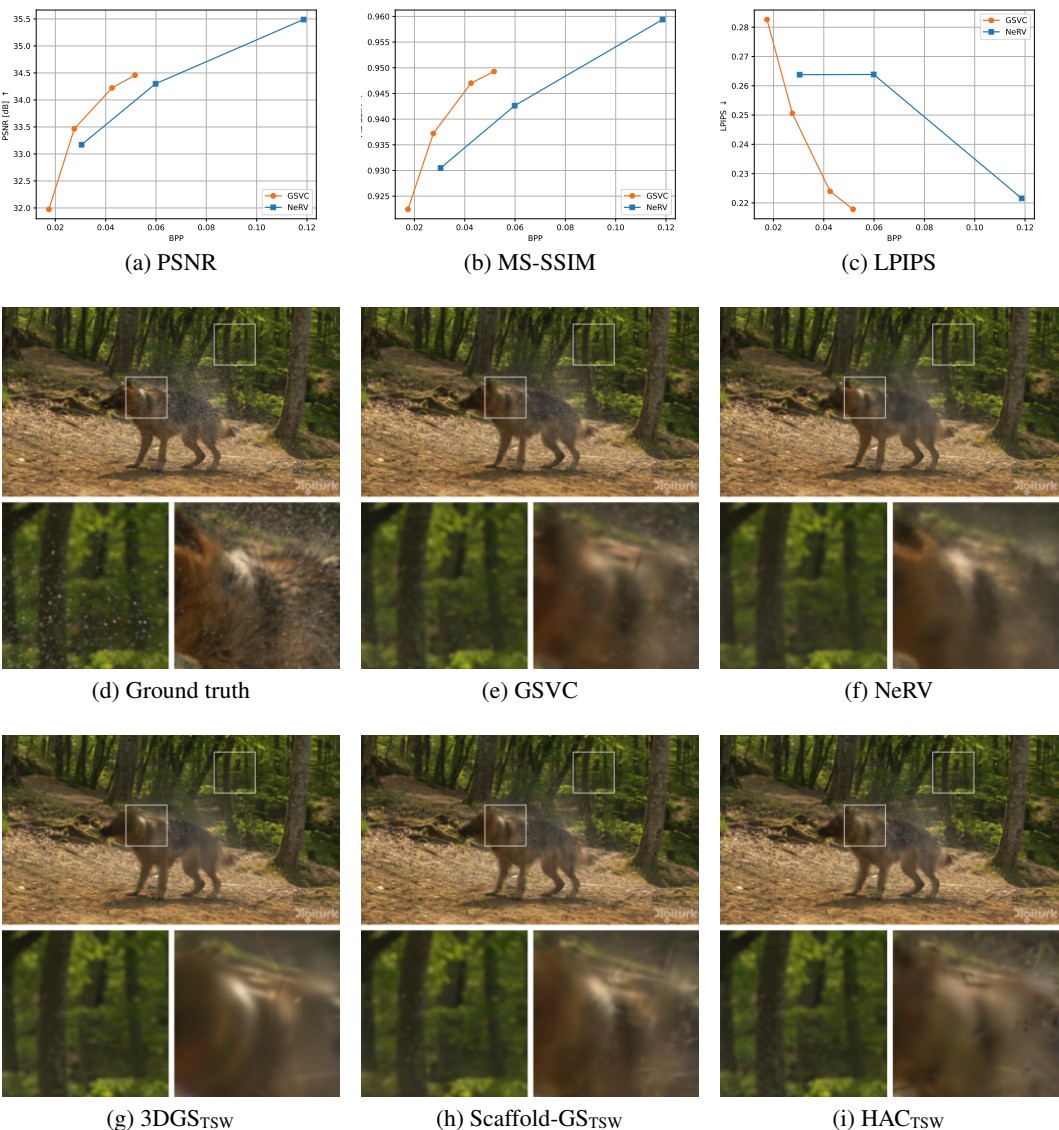

(a) PSNR

(b) MS-SSIM

(c) LPIPS

(d) Ground truth

(e) GSVC

(f) NeRV

(g) 3DGS$_{TSW}$

(h) Scaffold-GS$_{TSW}$

(i) HAC$_{TSW}$

Figure 15: ShakeNDry

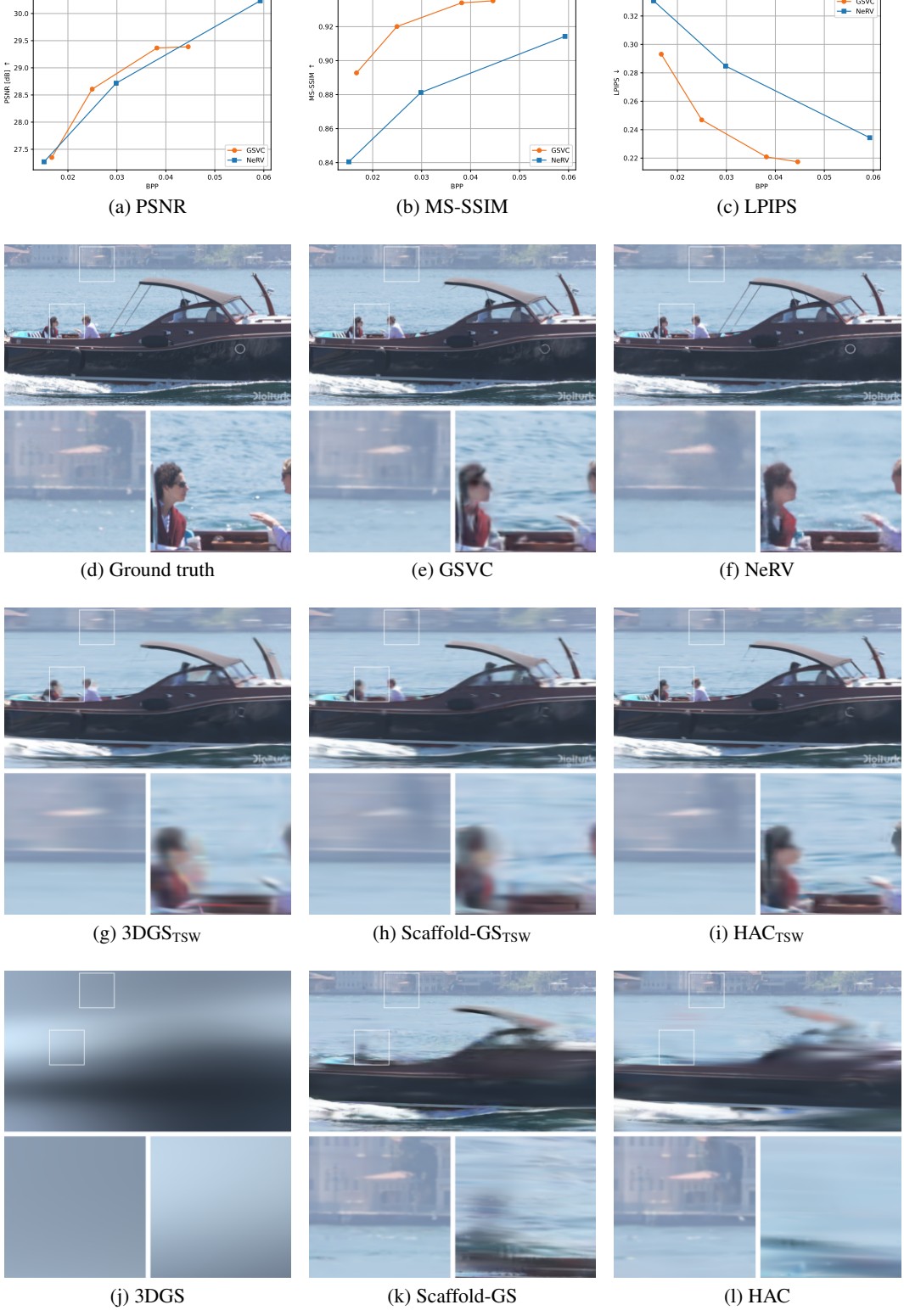

(a) PSNR

(b) MS-SSIM

(c) LPIPS

(d) Ground truth

(e) GSVC

(f) NeRV

(g) 3DGS$_{TSW}$

(h) Scaffold-GS$_{TSW}$

(i) HAC$_{TSW}$

(j) 3DGS

(k) Scaffold-GS

(l) HAC

Figure 16: YachtRide

