# OpenReview forum: "An Exploration with Entropy Constrained 3D Gaussians for 2D Video Compression"
_ICLR.cc/2025/Conference — ICLR 2025 Poster_

### Official Review · Reviewer_nojS · 2024-10-28

**Soundness:** 3
**Presentation:** 3
**Contribution:** 3
**Rating:** 6
**Confidence:** 5

**Summary:**

This paper proposes a new framework for video compression from the perspective of 3D Gaussian Splatting (3DGS). The key idea is to construct an interesting spatial-temporal modeling way called toast-like sliding window to cope with the 3D scene modeling in a short time period. This design enables the introduction of several 3DGS-based compression methods into video compression areas. With further careful designs like optical-flow supervision and time-aware Gaussians, the method achieves better compression quality then previous implicit-representation-based approaches like NeRV.

**Strengths:**

1. The idea of toast-like sliding windows is interesting. It provides an insightful design by treating the time step as an extra dimension to construct a 3D structure.
2. The overall writing is clear and well-motivated. The logical flow is very clear and easy to understand.
3. The overall idea of the exploration of video compression from the 3DGS perspective is novel.

**Weaknesses:**

1.  Why do toast-like sliding windows need to render a short future period of content? It would be better to provide more illustrations about the detailed rendering process.
2. Ablation study of FiLM design. The analysis of the change from MLP to FiLM-based network structure is missing.
3. It would be better to include a more comprehensive comparison of video compression approaches like H.264/265 and HNeRV for in-depth analysis.
4.  There are several typos in the writing like line 243, us-> use.

**Questions:**

1. It would be better to provide the rasterization process when applying the Orthographic Projection in the context of Gaussian Splatting. Including these changes in the appendix will help the readers understand the paper more clearly.
2. How the anchor points are selected in the toast-like sliding windows?

---

> ### Author Response · Authors · 2024-11-20
>
> We sincerely express our gratitude for dedicating your valuable time to providing insightful suggestions that can enhance our paper. Your praise regarding our novelty, methodology, and writing has greatly motivated us. Our detailed responses to all of your concerns are presented below.
>
>
>
> **W1:**
> Why do toast-like sliding windows need to render a short future period of content? It would be better to provide more illustrations about the detailed rendering process.
>
> **R1:**
> Thanks for your question. The idea of rendering a frame in bi-direction fashion is inspired by the B-frame in traditional codec, which utilizes the bi-directional similarity along time axis. We have added more descriptions in the revised version.
>
> **W2:**
> Ablation study of FiLM design. The analysis of the change from MLP to FiLM-based network structure is missing.
>
> **R2:**
> Thanks for your question. The results are shown in Table 2, w/o FiLM entry. We have added more descriptions to enhance the clarity.
>
> **W3:**
> It would be better to include a more comprehensive comparison of video compression approaches like H.264/265 and HNeRV for in-depth analysis.
>
> **R3:**
> Thanks for your suggestion. We have added the comparison with H.264, H.265 and HNeRV to the revised version. The experiments demonstrate that GSVC achieves comparable MS-SSIM and LPIPS performance with H.265 (veryslow) at low bit rate region and outperforms many INR-based methods. As an early work exploring the feasibility of 3D Gaussian Splatting in video compression, we believe this is a positive proof-of-concept.
>
> **W4:**
> There are several typos in the writing like line 243, us-> use.
>
> **R4:**
> Thanks for your comment. We have fixed typos in the revised version.
>
> **Q1:**
> It would be better to provide the rasterization process when applying the Orthographic Projection in the context of Gaussian Splatting. Including these changes in the appendix will help the readers understand the paper more clearly.
>
> **A1**
> Thanks for your suggestion. We will add more descriptions in the appendix as soon as possible.
>
> **Q2:**
> How the anchor points are selected in the toast-like sliding windows?
>
> **A2:**
> Thanks for your question. We select anchors according to their distance from the camera plane of being rendered frame. All anchors with distances smaller than $h$ will participate in the rendering.
>
>
> Thank you again for your thoughtful comments! If you have any further concerns, feel free to reach out to us.

---

> ### Author Response · Authors · 2024-11-24
> **Update**
>
> **Q1:**
> It would be better to provide the rasterization process when applying the Orthographic Projection in the context of Gaussian Splatting. Including these changes in the appendix will help the readers understand the paper more clearly.
>
> **A1-Updated:** We have added more figures and illustrations about the rendering process, please refer to Appendix A.3 in the revised version.   :)
>
> We appreciate your insightful feedback! Please contact us if you have any further questions.

---

### Official Review · Reviewer_jDPh · 2024-11-01

**Soundness:** 3
**Presentation:** 3
**Contribution:** 3
**Rating:** 6
**Confidence:** 4

**Summary:**

This paper explores 3DGS for video compression. It Introduces TSW projection to adapt 3D Gaussian Splatting for 2D video representation. This paper also develops an end-to-end trainable video compression method based on deformable Gaussian and optical guidance to better model motion content in videos.

**Strengths:**

1. Orthographic projection and sliding windows for videos are reasonable ideas for adapting the original perspective projection in 3DGS.

2. Experiments on the UVG dataset show that the developed GSVC codec achieves RD performance compared to NeRV and offers efficient rendering with inherent stream decoding capability. Besides, this paper presents a relatively complete work for the codec workflow.

3. The paper mentions that the code will be open-source. I appreciate the promise of open source, especially in video compression.

**Weaknesses:**

1. the proposed method still needs to train per video. Which i think is a limitation for real-world usage.

2. As video codecs are a well-developed method, this paper does not include comparisons to traditional codecs. I know neural representation for compression is a novel and hot feild. But it’s more valuable and practice to highlight the specific strengths of each methods, especially in terms of the computation costs for encoding.

**Questions:**

1. Figure 1's ground truth image seems to be with low quality. Based on my knowdege, 3DGS's reconstruct is prune to have high numerical results but lack of details. Have the author consider this issue for compresssion task ?
2. It is hard to see if there is a temporal inconsistency without video examples. Can you provide video comparisons ?

---

> ### Author Response · Authors · 2024-11-20
>
> We sincerely express our gratitude for dedicating your valuable time to providing insightful suggestions that can enhance our paper. Your appreciation regarding our novelty and completeness has greatly encouraged us. Our detailed responses to all of your concerns are presented below.
>
>
> **W1:**
> the proposed method still needs to train per video. Which i think is a limitation for real-world usage.
>
> **R1:**
> Thanks for your question. This is a common issue for INR/3DGS-based methods. Currently, we think this paradigm is suitable for a compress-once decode-often scenario, which is insensitive to encoding time. To solve the issue completely, we think there are two possible ways
> 1. Directly generate INR/3DGS from the input video, like MVSplat[1].
> 2. Improve the training procedure to reduce the encoding time.
>
>
>
> **W2:**
> As video codecs are a well-developed method, this paper does not include comparisons to traditional codecs. I know neural representation for compression is a novel and hot field. But it's more valuable and practice to highlight the specific strengths of each methods, especially in terms of the computation costs for encoding.
>
> **R2:**
> Thanks for your suggestion. We have added the comparison with H.264, H.265 to the revised version. The experiments demonstrate that GSVC achieves comparable MS-SSIM and LPIPS performance with H.265 (veryslow) at low bit rate region and outperforms many INR-based methods. As an early work exploring the feasibility of 3D Gaussian Splatting in video compression, we believe this is a positive proof-of-concept. We are conducting experiments to measure the encoding time and we will add the results as soon as possible.
>
> **Q1:**
> Figure 1's ground truth image seems to be with low quality. Based on my knowdege, 3DGS's reconstruct is prune to have high numerical results but lack of details. Have the author consider this issue for compresssion task ?
>
> **A1:**
> Thanks for your question. UVG dataset is widely used in many video compression research with the resolution of 1920x1080. The low quality may be due to the resize operation. We also demonstrate the comprehensive results of PSNR, MS-SSIM, and LPIPS to reduce the gap between visual quality and numerical results. In our experiments, we observe GSVC demonstrates better quantity and quality for regular moving objects like background in Bosphorus with sufficient details. Deteriorated details are found in fast-moving objects like legs in Jockey. The proposed components like FiLM are used to improve details. We demonstrate the effectiveness of these components in Table 2.
>
> **Q2:** It is hard to see if there is a temporal inconsistency without video examples. Can you provide video comparisons?
>
> **A2:** Thanks for your suggestion. We provide video comparisons in [anonymous url](https://anonymous.4open.science/r/GSVC-10656/readme.md). Please follow the readme to run the visualization.
>
> Thank you again for your insightful suggestion. If you have any further question, feel free contact us.
>
> [1] Chen, Yuedong, et al. "Mvsplat: Efficient 3d gaussian splatting from sparse multi-view images." European Conference on Computer Vision. Springer, Cham, 2025.

---

> ### Author Response · Authors · 2024-11-24
> **Update**
>
> **W2:**
> As video codecs are a well-developed method, this paper does not include comparisons to traditional codecs. I know neural representation for compression is a novel and hot field. But it's more valuable and practice to highlight the specific strengths of each methods, especially in terms of the computation costs for encoding.
>
> **R2-Updated:** Beside the results in the previous response, here is the encoding time of INR/3DGS methods.
> | Method | Frames | Time | GPU Util. |
> | :---: | :---: |:--- |:---: |
> | NeRV |  240k | 22ks (6h6m)  | 100%|
> | HNeRV |  180k | 13.8ks\~28.2ks (3h50m\~7h50m) | 100%|
> | GSVC |  **80k** |   **12.4ks (3h26m)** | 85% |
>
> *Frames* represents the number of total frames rendered during training. We show the results of a typical 600-frame video in UVG dataset. The training time of HNeRV is sensitive to model size *i.e.* bpp, so we demonstrate a range here. *GPU Util.* is from nvidia-smi. GSVC outperforms other INR-based methods. Since there is no accepted metric like FLOPS to measure the computational complexity of 3D Gaussian Splatting rendering, we use GPU utilization from nvidia-smi as a reference. A utilization below 100% indicates that GSVC's computational consumption is lower than other methods. However, it also suggests that the GPU's capability is not fully tapped. This limitation calls for further exploration and improvement in the future.
>
> We have added the result to Appendix A.4 in the revised version.   :)
>
>
> We appreciate your valuable comments! Please don't hesitate to tell us if you have any further concerns.

---

### Official Review · Reviewer_JGLW · 2024-11-02

**Soundness:** 3
**Presentation:** 3
**Contribution:** 3
**Rating:** 8
**Confidence:** 5

**Summary:**

The authors propose a novel approach to 2D video compression using 3D Gaussian Splatting (3DGS). They represent 2D video sequences as a volumetric distribution in 3D space, using X-Y-T coordinates. To reduce temporal redundancy, they introduce a Toast-like Sliding Window (TSW) projection method that considers Gaussian points within a temporal window centered on the rendering plane. The framework incorporates optical flow information and a FiLM-based deformation field to better capture temporal dynamics. An entropy coding scheme is implemented to optimize model compression and enhance streaming efficiency.

The paper presents three main contributions:
* Introduction of a novel paradigm that represents 2D video using 3D Gaussian models, supported by the TSW projection technique
* Development of GSVC, a comprehensive compression pipeline that combines HAC, deformable fields, and opacity flow regularization
* Experimental validation showing GSVC achieves comparable quality to NeRV while providing 30% faster rendering and flexible bitrate-availability tradeoffs

**Strengths:**

* Pioneering application of 3DGS to video compression
* Minimal assumptions about video content, relying primarily on temporal continuity, which enables broad applicability
* High adaptability with various 3DGS models, opening promising research directions
* Demonstrated practical utility through flexible bitrate-availability tradeoffs, particularly valuable for streaming applications

**Weaknesses:**

I do really like the idea of the paper yet the evaluation section could be further polished.
* Limited comparison baseline, with NeRV as the only reference. The evaluation would benefit from comparisons with other neural video compression methods
* Evaluation only focus on low bitrate/bpp scenarios, while contemporary neural compression methods (e.g., VCT, DCVC) typically demonstrate performance at higher bitrates with PSNR values exceeding 34 on the UVG dataset.

**Questions:**

- In Figure 5, there remains a gap in frame availability even after complete bit decoding. What factors contribute to this limitation?
- Another concern is about the depth sorting when it comes to rendering. there could be a scenario when two gaussians (lets say Ga and Gb), in the frame t-1, fall into Vf, while in the next frame t, they fall into Vb, due to the time plane moving forward. Yet in this case the relative depths of Ga and Gb to the image plane are reversed
  - for example, in t-1, Ga, Gb in Vf, and Ga is close to the plane, and Gb is further to the plane, then in t, Ga, Gb in Vb, it must be that Ga is further to the plane and Gb looks closer to the plane
  - Such kind of reversing might lead to totally different coloring of the same pixel after alpha blending. has such inconsistency in consecutive frames ever been observed in your experiments?
  - Also how the two images projected from Vb and Vf get mixed into a single image? simply averaging them?
  - It would be interesting if you could visualize the xyt space as a normal 3D space and in the meantime, visualize the rendering of one of the frames from nearby Gaussians. That will be cool.

---

> ### Author Response · Authors · 2024-11-20
>
> We sincerely express our gratitude for dedicating your valuable time to providing insightful suggestions and positive feedback on our work. Your appreciation and acknowledgment of our manuscripts have greatly inspired us. Our detailed responses to all your questions are provided below.

---

> ### Author Response · Authors · 2024-11-20
>
> **W1&W2:** Limited comparison baseline, with NeRV as the only reference. The evaluation would benefit from comparisons with other neural video compression methods. Evaluation only focus on low bitrate/bpp scenarios, while contemporary neural compression methods (e.g., VCT, DCVC) typically demonstrate performance at higher bitrates with PSNR values exceeding 34 on the UVG dataset.
>
> **R:**
> Thanks for your suggestion. We have added the comparison with VCT and DCVC. The proposed GSVC has a performance gap between contemporary neural compression methods. In high-bitrate regions, the performance gap becomes larger, which occurs in similar early INR/3DGS works as well[1]. However, our method provides a real-time rendering speed, which is necessary in many practical scenarios. As an early work exploring the feasibility of 3D Gaussian Splatting in video compression, we believe the work opens promising research directions in video compression and is worth further exploration.
>
>
> **Q1:** In Figure 5, there remains a gap in frame availability even after complete bit decoding. What factors contribute to this limitation?
>
> **A1:**
> Thanks for your question. When the bit stream is completely decoded, all frames are available for rendering for both GSVC and NeRV. The gap is due to the overlapped border of the figure. We have refined the figure in the revised version to enhance the clarity.
>
> **Q2.1**
> Another concern is about the depth sorting when it comes to rendering. there could be a scenario when two gaussians (lets say Ga and Gb), in the frame t-1, fall into Vf, while in the next frame t, they fall into Vb, due to the time plane moving forward. Yet in this case the relative depths of Ga and Gb to the image plane are reversed
>
> for example, in t-1, Ga, Gb in Vf, and Ga is close to the plane, and Gb is further to the plane, then in t, Ga, Gb in Vb, it must be that Ga is further to the plane and Gb looks closer to the plane
> Such kind of reversing might lead to totally different coloring of the same pixel after alpha blending. has such inconsistency in consecutive frames ever been observed in your experiments?
>
> **A2.1**
> Thanks for your question. The inconsistency of color is not significant. We think the reason is twofold. First, the color of each Gaussian point is generated by the corresponding network, which takes feature $\mathcal{A}$ and time positional encoding $\mathrm{pe}$ as input.  $\mathcal{A}$ is time independent but  $\mathrm{pe}$  is time dependent. The network has sufficient information to identify this reversion. Second, some inconsistency in vanilla 3DGS occurs in novel view synthesis, which is not a problem in video compression, since the network will see all frames at encoding *i.e.* training stage.
> Different from color, we find more inconsistency occurs in highly dynamic objects, which is a valuable direction to be explored in future work.
>
> **Q2.2**
> Also how the two images projected from Vb and Vf get mixed into a single image? simply averaging them?
>
> **A2.2**
> Thanks for your question. Currently we simply average them.
>
> **Q2.3**
> It would be interesting if you could visualize the xyt space as a normal 3D space and in the meantime, visualize the rendering of one of the frames from nearby Gaussians. That will be cool.
>
> **A2.3**
> Thanks for your suggestion. We will add the visualization of the progressive rendering of a frame to the revised version as soon as possible. We also provide an example of an interactive visualization of anchor points in xyt space in [anonymous url](https://anonymous.4open.science/r/GSVC-10656/readme.md). Please follow the readme to run the visualization. Besides, we will try to visualize the rendering process in 3D space.
>
>
> Thank you again for your valuable feedback! If you have any further questions or suggestions,
> please don't hesitate to tell us.
>
> [1] Zhang, Xinjie, et al. "Gaussianimage: 1000 fps image representation and compression by 2d gaussian splatting." European Conference on Computer Vision. Springer, Cham, 2025.

---

> ### Author Response · Authors · 2024-11-24
> **Update**
>
> **Q2.3:**
> It would be interesting if you could visualize the xyt space as a normal 3D space and in the meantime, visualize the rendering of one of the frames from nearby Gaussians. That will be cool.
>
> **A2.3-Updated:** We have added more visualization about the rendering process, please refer to Appendix A.3 in the revised version.  :)
>
> We appreciate your thoughtful suggestion! Please feel free to reach out if you have any more concerns.

---

> ### Comment · Reviewer_JGLW · 2024-11-24
>
> not too bad for ssim, good paper. will raise the score. yet HAC has its own alternative for torchac (https://github.com/YihangChen-ee/HAC?tab=readme-ov-file#updates), i didnt dive into their code, can you elaborate whats new for your codc compared with HAC's?

---

> ### Author Response · Authors · 2024-11-25
>
> We appreciate your acknowledgment of our work!
>
> One of the reasons to implement our entropy codec is that we developed GSVC based on an early version of HAC, at that time the alternative codec was unavailable and torchac was slow.
>
> There are two differences between our codec and arithmetic codec in HAC (noted as arithmetic_cuda):
> 1. Our codec is based on ANS theory, while arithmetic_cude is based on arithmetic coding theory. In general, ANS codec is faster than arithmetic codec at the cost of a little higher overhead ($<1$%) [1].
> 2. Our codec utilizes abundant threads in GPU  by parallel scanning the whole support set simultaneously when decoding the next symbol. Arithmetic_cuda finds the correct symbol by binary search and utilizes the parallel capability of GPU by chunking.
>
> Following is the comparison of different methods on a synthetic message with 20000 symbols. Limited by the number of threads in one block of GPU, our method can scan 256 or 512 entries in a support set (depending on the architecture of GPU) at a time. When the support set is very large, the strategy will degrade to linear scan. As illustrated in Table 1, our method is slower than arithmetic_cuda. When the support size is small *i.e.* smaller than 256 or 512, our decoding is faster than others, as shown in Table 2.
>
> **Table 1: Support set size 30000**
> | Method | Encoding (s) | Decoding (s)|
> | :--- | :--- |:--- |
> | torchac |  1.733 | 2.070 |
> | arithmetic_cuda |  0.021 | **0.046** |
> | Ours | **0.005** |   0.243 |
>
> **Table 2: Support set size 300**
> | Method | Encoding (s) | Decoding (s) |
> | :--- | :--- |:--- |
> | torchac |  0.017 | 0.019 |
> | arithmetic_cuda |  0.014 | 0.032 |
> | Ours | **0.005** |  **0.012** |
>
> It should be noted that our codec can be further enhanced by chunking like arithmetic_cuda. Cooperating binary search and parallel scanning is another feasible way to boost performance.
>
> Thanks for your question! Please tell us if you have any further concerns. :)
>
> [1] Bamler, Robert. "Understanding entropy coding with asymmetric numeral systems (ans): a statistician's perspective." arXiv preprint arXiv:2201.01741 (2022).

---

### Official Review · Reviewer_czLV · 2024-11-03

**Soundness:** 2
**Presentation:** 3
**Contribution:** 2
**Rating:** 5
**Confidence:** 5

**Summary:**

This paper introduces a 2D video compression method using 3D Gaussian Splatting (3DGS), aiming for faster frame reconstruction and stream decoding. However, concerns remain about its limited novelty, as many techniques (e.g., time embeddings, Gaussian attribute adjustments) are adapted from NeRF and 4DGS. The framework largely assembles prior methods (e.g., HAC, GPCC, and optical flow loss from 4DGS) without substantial innovation, raising questions about the applicability and originality of the approach in RGB video compression.

**Strengths:**

1. Improved Inference Speed:

The compression framework outperforms INR-based inference in rendering speeds, a notable improvement in practical applications where real-time decoding is essential.

2. Broad Performance Evaluation:

The authors provide comprehensive experiments and comparisons with alternative compression methods, demonstrating that GSVC achieves favorable rate-distortion performance across different quality metrics.

**Weaknesses:**

1. Lack of Novelty:

The technical novelty of the work is limited. Key techniques, such as time-position embeddings and Gaussian attribute variation over time, have been previously introduced in NeRF and 4DGS [1]. The approach of offsetting Gaussian positions based on features at different timestamps is conceptually similar to the offset extraction in dynamic 4DGS methods. The overall compression framework aligns closely with HAC [2], while the compression of x,y,z coordinates with GPCC was introduced in other prior work. Similarly, optical flow loss is a component already implemented in 4DGS [3]. And the GPU-based ANS ae/ad is already implemented in CompressAI[4].


2. No Comparison with HEVC/VVC:

The paper lacks a direct performance comparison with widely-used standards like HEVC or VVC, making it difficult to gauge its practical effectiveness against traditional video compression methods.


3. Questionable Value for INR/3DGS in RGB Video Compression:

The application of INR/3DGS techniques to RGB video compression seems inherently misaligned with the goals of video compression. Compression frameworks for RGB video typically rely on extensive data to learn a generalized data distribution, leveraging priors that do not exist within a single video or image.

[1] 4D Gaussian Splatting for Real-Time Dynamic Scene Rendering

[2] HAC: Hash-grid assisted context for 3D Gaussian splatting compression

[3] MotionGS: Exploring Explicit Motion Guidance for Deformable 3D Gaussian Splatting

[4] https://github.com/InterDigitalInc/CompressAI

**Questions:**

Considering the typical aim of RGB image/video compression to leverage extensive priors learned from large datasets, how does the proposed framework add value when applied to compress a single video? Is there any particular benefit in this context?

---

> ### Author Response · Authors · 2024-11-20
>
> We sincerely thank you for your precious time and effort in providing a wealth of suggestions to enhance the quality of our paper. We have carefully read all the comments and provide detailed point-by-point responses as follows. Hopefully, we can adequately address your concerns.

---

> ### Author Response · Authors · 2024-11-20
>
> **W1:** Lack of Novelty: The technical novelty of the work is limited. Key techniques, such as time-position embeddings and Gaussian attribute variation over time, have been previously introduced in NeRF and 4DGS [1]. The approach of offsetting Gaussian positions based on features at different timestamps is conceptually similar to the offset extraction in dynamic 4DGS methods. The overall compression framework aligns closely with HAC [2], while the compression of x,y,z coordinates with GPCC was introduced in other prior work. Similarly, optical flow loss is a component already implemented in 4DGS [3]. And the GPU-based ANS ae/ad is already implemented in CompressAI[4].
>
> **R1:**
> Thanks for your comment. The main contribution of the manuscript is the **Toast-like Sliding Window Orthographic Projection (TSW)**, which effectively transforms any 3DGS family methods into a video representation model. This is the first attempt to apply 3D Gaussian Splatting to video compression. We detail the mechanism in Section 3.2 and present the experiment results in Section 4.2 and Table 1. The mentioned techniques from the literature are part of the GSVC, but we tailored these methods to accommodate the compression task. The comparison of the original methods and usages in our work are as follows:
>
> 1. *Time-position embeddings and Gaussian attribute variation over time, have been previously introduced in NeRF and 4DGS [1]. The approach of offsetting Gaussian positions based on features at different timestamps is conceptually similar to the offset extraction in dynamic 4DGS methods.*
>
>     Positional encoding or time-position embedding is a widely used trick and we follow the common usage in previous works. For Gaussian attribute variation over time, we propose our deformation network based on Deformable 3DGS [11], which outperforms 4DGS. Since the deformation network in Deformable 3DGS is too large to be used in compression tasks, we propose to use FiLM [12] structure to create a parameter-efficient network, which has never been explored in previous work as far as we know. The ablation of the components is presented in Table 2 in Section 4.4.
>
> 2. *The overall compression framework aligns closely with HAC.*
>
>     The main difference between GSVC and HAC originates from the design of TSW. Since the culling operation in TSW is isolated from the rendering process, different from the mixed procedure of culling and rendering in common perspective projection in 3DGS, our framework can easily achieve stream decoding, which is practical in many realistic scenarios. We demonstrate the experiment results in Fig. 5b.
>
> 3. *The compression of x,y,z coordinates with GPCC was introduced in other prior work.*
>
>     Few of the previous 3DGS compression work employs the GPCC in the pipeline. One of the possible reasons is that the quantized coordinates only occupy a small part of the encoded model [2]. However, for video compression tasks, the size of plain quantized coordinates is nonnegligible. As a result, we introduce GPCC to improve RD performance. The ablation of the components is presented in Table 2 in Section 4.4.
>
> 4. *Optical flow loss is a component already implemented in 4DGS (MotionGS)[3].*
>
>     **The mentioned 4DGS (MotionGS) is submitted to Arxiv on 10 Oct 2024, which is latter than the submission deadline of ICLR 2025**. We believe that the reference paper and our usage of optical flow loss should be considered as contemporaneous work.
>
> 5. *GPU-based ANS ae/ad is already implemented in CompressAI [4]*
>
>     CompressAI only provides a CPU-based ANS codec implementation. Besides, decoding in a probing fashion which utilizes the large number of threads in GPU has not been investigated in previous work.

---

> ### Author Response · Authors · 2024-11-20
>
> **W2:**
> No Comparison with HEVC/VVC:
> The paper lacks a direct performance comparison with widely-used standards like HEVC or VVC, making it difficult to gauge its practical effectiveness against traditional video compression methods.
>
> **R2:**
> Thanks for your suggestion. We have added the comparison with H.265/HEVC to the revised version. The experiments demonstrate that GSVC achieves comparable MS-SSIM and LPIPS performance with H.265 (veryslow) at low bit rate region and outperforms many INR-based methods. As an early work exploring the feasibility of 3D Gaussian Splatting in video compression, we believe this is a positive proof-of-concept.
>
> **W3:**
> Questionable Value for INR/3DGS in RGB Video Compression:
> The application of INR/3DGS techniques to RGB video compression seems inherently misaligned with the goals of video compression. Compression frameworks for RGB video typically rely on extensive data to learn a generalized data distribution, leveraging priors that do not exist within a single video or image.
>
> **R3:**
> Thanks for your valuable question. We would like to illustrate the value of INR/3DGS compression paradigm from three perspectives.
>
> 1. Data distribution within a single video/image cooperating with inductive bias from model structure is sufficient for some compression task.
>
>     Data distribution is critical for codec to achieve efficient compression. Learning data distribution from extensive data is a very effective way, but this data-driven method is not the only way to attain the distribution required by low-level tasks like compression. A learning-based model can easily learn the temporal similarity in a single video or spatial adjacent similarity in a single frame. Some INR-based image compression methods outperform HEVC in many benchmarks [5]. Another important component of the INR/3DGS paradigm is inductive bias from model structure. Traditional codecs reduce the redundancy by hand-crafted rules. For example, most video codecs include a motion estimation module. Designs like motion estimation rely on the human's observation of video data i.e. the data priors learning by humans. Many INR/3DGS methods utilize these priors in the design of model structures to improve performance [5, 6]. For many compression tasks, especially when extreme RD performance is unnecessary but practical metrics like rendering speed are important, INR/3DGS compressing is very useful. Some recent works also demonstrate the effectiveness of data distribution within a single video/image and inductive bias from model structure [5].
>
> 2. The INR/3DGS method is orthogonal to data-driven method. It is possible to utilize extensive data in INR/3DGS paradigm.
>
>     In this manuscript, we present a video compression method learning from a single video. However, the design choice does not necessarily mean we cannot build an INR/3DGS compression method capable of leveraging extensive data. For INR, numerous works have investigated improving performance by methods like meta-learning [7]. For 3DGS, MVSplat [8] generates 3DGS directly from input photos through a pre-trained network, instead of learning from scratch. All these works learn a generalized data distribution in different manners. Although integrating these techniques with compression is non-trivial, they indicate the possibility of utilizing extensive data in the INR/3DGS paradigm and provide a promising direction to explore.
>
> 3. The INR/3DGS method provides some practical features in compression task.
>
>     From a more practical perspective, many end-to-end video compression methods require high-performance hardware. For example, VCT [9] achieves superior RD performance at the cost of only 1.4 FPS on TPUv4, which is far from practical usage. On the contrary, some INR-based methods with comparable RD performance have ~20 FPS rendering speed on A100 [10]. The proposed GSVC also achieves ~60 FPS on RTX 3090. For a compress-once decode-often scenario, the INR/3DGS paradigm is more useful.
>
> **Q1:** Considering the typical aim of RGB image/video compression to leverage extensive priors learned from large datasets, how does the proposed framework add value when applied to compress a single video? Is there any particular benefit in this context?
>
> **A1:** Thanks for your valuable question. We summarize the reasons in **R3** above.

---

> ### Author Response · Authors · 2024-11-20
>
> We hope these responses can address your concerns. Once again, we deeply appreciate your valuable suggestions for improving our work and would be delighted to further discuss with you.

---

> ### Author Response · Authors · 2024-11-20
>
> [1] Wu, Guanjun, et al. "4d gaussian splatting for real-time dynamic scene rendering." Proceedings of the IEEE/CVF Conference on Computer Vision and Pattern Recognition. 2024.
>
> [2] Chen, Yihang, et al. "Hac: Hash-grid assisted context for 3d gaussian splatting compression." European Conference on Computer Vision. Springer, Cham, 2025.
>
> [3] Zhu, Ruijie, et al. "MotionGS: Exploring Explicit Motion Guidance for Deformable 3D Gaussian Splatting." arXiv preprint arXiv:2410.07707 (2024).
>
> [4] https://github.com/InterDigitalInc/CompressAI
>
> [5] Kim, Hyunjik, et al. "C3: High-performance and low-complexity neural compression from a single image or video." Proceedings of the IEEE/CVF Conference on Computer Vision and Pattern Recognition. 2024.
>
> [6] Ladune, Théo, et al. "Cool-chic: Coordinate-based low complexity hierarchical image codec." Proceedings of the IEEE/CVF International Conference on Computer Vision. 2023.
>
> [7] Dupont, Emilien, et al. "Coin++: Data agnostic neural compression." arXiv preprint arXiv:2201.12904 1.2 (2022): 4.
>
> [8] Chen, Yuedong, et al. "Mvsplat: Efficient 3d gaussian splatting from sparse multi-view images." European Conference on Computer Vision. Springer, Cham, 2025.
>
> [9] Mentzer, Fabian, et al. "Vct: A video compression transformer." arXiv preprint arXiv:2206.07307 (2022).
>
> [10] Kwan, Ho Man, et al. "Hinerv: Video compression with hierarchical encoding-based neural representation." Advances in Neural Information Processing Systems 36 (2024).
>
> [11] Yang, Ziyi, et al. "Deformable 3d gaussians for high-fidelity monocular dynamic scene reconstruction." Proceedings of the IEEE/CVF Conference on Computer Vision and Pattern Recognition. 2024.
>
> [12] Perez, Ethan, et al. "Film: Visual reasoning with a general conditioning layer." Proceedings of the AAAI conference on artificial intelligence. Vol. 32. No. 1. 2018.

---

> > ### Comment · Reviewer_czLV · 2024-11-26
> >
> > Thank you for your detailed rebuttal. I appreciate the effort in addressing the comments and clarifying your contributions. However, I would like to clarify the following points regarding the novelty and methodology:
> >
> > 1. **Toast-like Sliding Window Orthographic Projection (TSW)**: The proposed TSW lacks novelty as it seems to be a straightforward reduction of the dimensions in prior work. Specifically, the 4DRealTime approach models Gaussian splats in the xyzt domain (4D), and removing the z-dimension results in an xyt (3D) Gaussian representation, which aligns closely with your method. This simplification does not constitute a significant contribution to the field.
> >
> > 2. **GPU-based ANS Implementation**: While the GPU-based ANS parallelizes the computation of PDF and CDF, ANS itself is inherently sequential. Similar GPU implementations have been explored in earlier PRs within MMCV and discussed in standardization forums, making this aspect less innovative.
> >
> > 3. **Lack of Targeted Experimental Results**: The claim that implicit and explicit representations perform better in specific scenarios is not substantiated with experimental results. A direct comparison in such scenarios (e.g., NIC/NVC benchmarks) is necessary to validate this argument.
> >
> > While I acknowledge the value of your work as an exploratory step, these points limit its perceived novelty and impact. My score remains consistent with those of other reviewers. I encourage you to strengthen these aspects in future work.

---

> ### Author Response · Authors · 2024-11-27
>
> We would like to thank the reviewer for your appreciation of our rebuttal and for providing insightful comments! We address the comments raised in the review point-by-point below:
>
>
> **Q1:** Toast-like Sliding Window Orthographic Projection (TSW): The proposed TSW lacks novelty as it seems to be a straightforward reduction of the dimensions in prior work. Specifically, the 4DRealTime approach models Gaussian splats in the xyzt domain (4D), and removing the z-dimension results in an xyt (3D) Gaussian representation, which aligns closely with your method. This simplification does not constitute a significant contribution to the field.
>
> **A1:** Thanks for your question. **To the best of our knowledge, there has been no research exploring the use of 3D Gaussian space in a sliding window manner, which utilizes the temporal similarity of video with a minimum assumption about video content**. Could you provide a reference of 4DRealTime, which removes the z-dimension results in an xyt (3D) Gaussian representation? Increasing the dimension of xyz space to xyzt space and reducing xyz space to xy space are both non-trivial problems and have their diversified explorations [1, 2, 3, 4, 5, 6]. Similarly, the proposed TSW mechanism is a novel but not the only way to represent 2D video in 3DGS paradigm. For example, we can represent 2D video by a series of frames in 2D Gaussian splating like GaussianImage [6], which increases xy space to xyt space. Another way is to decompose 3D Gaussians to 2D Gaussians and 1D Gaussians instead of decomposing 4D Gaussians to 3D Gaussians and 1D Gaussians [4], which aligns more closely to a straightforward reduction of the dimensions and is different from our method that directly projects time-variant 3D Gaussians in a sliding window to obtain one frame. These **unexplored ideas** all have their advantages and disadvantages, and can also benefit from the method proposed in this work. We look forward to more research focusing on this field. We hope we can address your concerns about novelty and clarify the differences between our method and previous works in the last response **R1** and this response.
>
> **Q2:** GPU-based ANS Implementation: While the GPU-based ANS parallelizes the computation of PDF and CDF, ANS itself is inherently sequential. Similar GPU implementations have been explored in earlier PRs within MMCV and discussed in standardization forums, making this aspect less innovative.
>
> **A2:** Thanks for your question. ANS itself is inherently sequential does not mean each step of encoding/decoding can not be parallelized. Except for computing PDFs and CDFs, the decoding procedure involves finding the correct symbol in the support set. The search process could be finished by linear scan or binary search. Benefiting from abundant threads in GPU, we can finish the linear scan in a parallel fashion at a time for a small support set (< 512). **This strategy has never been investigated before to the best of our knowledge.** Since the strategy will degrade to a faster linear scan for a very large support set, we can combine binary search and parallel scan to further boost performance in the future, like sort operation in C++ STL which combines merge sort and quick sort. Another important thing is that a single GPU thread is slower than a CPU thread, which means it is necessary to chunk the input message to exploit the parallel capability of GPU. We will consider the improvements as part of our future work. Besides, this is not a major contribution of our work. We developed the codec due to the lack of a fast entropy codec. Compared to adapting an unmerged feature in MMCV for our pipeline, implementing our own codec is easier.
>
>
> **Q3:** Lack of Targeted Experimental Results: The claim that implicit and explicit representations perform better in specific scenarios is not substantiated with experimental results. A direct comparison in such scenarios (e.g., NIC/NVC benchmarks) is necessary to validate this argument.
>
> **A3:** Thanks for your question. Some previous works support our claim. COOL-CHIC(ICCV23) [7] provides competitive RD performance of image compression with HEVC and hyperprior [8] with **less than 1% of hyperprior's computational complexity**, which is important for low-power devices like cell phones. C3(CVPR24) [9] further improves the RD performance of image compression and **significantly outperforms BPG/HEVC**. GaussianImage(ECCV24) [4] offers nearly **1000FPS decoding speed**. All these works demonstrate INR/3DGS paradigm is an avenue with great potential and worthy to explore in the future. We hope that you will grasp this promising direction, rather than overlook it due to a lack of understanding.
>
> We hope these replies help to clarify your concerns. We truly appreciate your constructive suggestions for improving our work and would be happy to discuss this with you further.

---

> ### Author Response · Authors · 2024-11-27
>
> [1] Wu, Guanjun, et al. "4d gaussian splatting for real-time dynamic scene rendering." Proceedings of the IEEE/CVF Conference on Computer Vision and Pattern Recognition. 2024.
>
> [2] Zhu, Ruijie, et al. "MotionGS: Exploring Explicit Motion Guidance for Deformable 3D Gaussian Splatting." arXiv preprint arXiv:2410.07707 (2024).
>
> [3] Yang, Ziyi, et al. "Deformable 3d gaussians for high-fidelity monocular dynamic scene reconstruction." Proceedings of the IEEE/CVF Conference on Computer Vision and Pattern Recognition. 2024.
>
> [4] Yang, Zeyu, et al. "Real-time photorealistic dynamic scene representation and rendering with 4d gaussian splatting." arXiv preprint arXiv:2310.10642 (2023).
>
> [5] Lin, Youtian, et al. "Gaussian-flow: 4d reconstruction with dynamic 3d gaussian particle." Proceedings of the IEEE/CVF Conference on Computer Vision and Pattern Recognition. 2024.
>
> [6] Zhang, Xinjie, et al. "Gaussianimage: 1000 fps image representation and compression by 2d gaussian splatting." European Conference on Computer Vision. Springer, Cham, 2025.
>
> [7] Ladune, Théo, et al. "Cool-chic: Coordinate-based low complexity hierarchical image codec." Proceedings of the IEEE/CVF International Conference on Computer Vision. 2023.
>
> [8] Ballé, Johannes, et al. "Variational image compression with a scale hyperprior." arXiv preprint arXiv:1802.01436 (2018).
>
> [9] Kim, Hyunjik, et al. "C3: High-performance and low-complexity neural compression from a single image or video." Proceedings of the IEEE/CVF Conference on Computer Vision and Pattern Recognition. 2024.

---

### Author Response · Authors · 2024-11-20
**Common Concern**

We sincerely appreciate all the reviewers for dedicating their valuable time and effort to review our paper and provide insightful comments and suggestions. Encouragingly, reviewers praise the GSVC's novelty and reasonableness ( R#JGLW, R#jDPh, R#nojS ), completeness (R#JGLW, R#jDPh), high adaptability (R#JGLW ) and appreciate the writing ( R#nojS) and broad evaluation (R#czLV) of our work.

We would like to express our sincerest gratitude once more for your invaluable time and efforts in evaluating our work! A revision of our paper that incorporates your precious suggestions will soon be uploaded to further improve the paper's quality.

---

### Meta-Review · Area_Chair_iDAU · 2024-12-23

**Metareview:**

The paper leverages 3D Gaussian Splatting (3DGS) for video compression, introducing the Toast-like Sliding Window (TSW) for efficient 2D video representation. The proposed GSVC framework uses deformable Gaussians and optical flow to improve temporal modeling and reduce redundancy. GSVC outperforms NeRV in rate-distortion and achieves 30–40% faster frame reconstruction, offering a practical and efficient solution for video compression.

Strengths:
Novelty: First to apply 3D Gaussian Splatting (3DGS) to video compression with innovative temporal modeling via Toast-like Sliding Window (TSW).
Efficiency: Achieves faster rendering and stream decoding, outperforming INR-based methods.
Robust Performance: Shows strong rate-distortion performance and adaptability through comprehensive evaluations.
Clear Writing and Open Source: Well-presented work with a commitment to releasing open-source code.

Weaknesses:
Limited Novelty: Adapts existing techniques from NeRF, 4DGS, and HAC, offering minimal originality.
Lacking Comparisons: Missing benchmarks against HEVC/VVC and advanced neural methods like VCT and DCVC.
Per-Video Training: Requires per-video training, reducing practicality for real-world use.
Evaluation Gaps: Limited focus on high-bitrate scenarios and missing ablations on FiLM design and rendering details.

The paper received divergent reviews.
We weakly accept the paper because it is the first attempt of using 3DGS on video compression task. But we would like the authors to include limitation section in the paper to detail the weaknesses of the method and also release the code as promised.

**Additional Comments On Reviewer Discussion:**

Reviewer feedback on the paper was mixed, with key concerns and responses outlined below:

Lack of Novelty and Relevance (Reviewer czLV):
Concern: Questioned the originality of the approach, viewing GSVC as a simplified 4DGS with limited value for RGB video compression.
Response: The rebuttal did not fully address these concerns, and the reviewer maintained a negative score.

Support for Contribution (Reviewer JGLW):
Positive View: Emphasized the significant effort behind adapting 4DGS for 2D video compression through TSW, HAC pipeline, FiLM, and customized entropy codecs. JGLW argued that GSVC reduces temporal redundancy effectively and demonstrates practical value, acknowledging its contribution despite its perceived simplicity.

Overall, the discussion highlights a divide between skepticism regarding novelty and appreciation of the practical efforts and results. These mixed perspectives weigh toward cautious acceptance, particularly given the method's demonstrated utility.

---

### Decision · Program_Chairs · 2025-01-22

Accept (Poster)